# WHEN DO TRANSFORMERS LEARN HEURISTICS FOR GRAPH CONNECTIVITY?

## ABSTRACT

Transformers often fail to learn generalizable algorithms, instead relying on brittle heuristics. Using graph connectivity as a testbed, we explain this phenomenon both theoretically and empirically. We consider a simplified Transformer architecture, the disentangled Transformer, and prove that an $L$-layer model has capacity to solve for graphs with diameters up to exactly $3^L$, implementing an algorithm equivalent to computing powers of the adjacency matrix. We analyze the training-dynamics, and show that the learned strategy hinges on whether most training instances are within this model capacity. Within-capacity graphs (diameter $\leq 3^L$) drive the learning of a correct algorithmic solution while beyond-capacity graphs drive the learning of a simple heuristic based on node degrees. Finally, we empirically demonstrate that restricting training data within a model's capacity leads to both standard and disentangled transformers learning the exact algorithm rather than the degree-based heuristic.

## 1 INTRODUCTION

Large language models (LLMs) based on the Transformer architecture have demonstrated remarkable capabilities, yet their success is often shadowed by failures on tasks that demand robust, algorithmic reasoning. A growing body of evidence shows that, instead of learning generalizable algorithms, these models frequently rely on brittle shortcuts and spurious correlations that exploit statistical cues in the training data (Niven & Kao, 2019; Geirhos et al., 2020; Tang et al., 2023; Yuan et al., 2024; Zhou et al., 2024b; Ye et al., 2024). This shortcut reliance contributes to poor out-of-distribution (OOD) generalization, vulnerability to adversarial prompts, and unreliability on multi-step reasoning tasks (Zou et al., 2023; Deng et al., 2024; Li et al., 2024). Evidence spans domains: in natural language inference, models pick up lexical-overlap heuristics rather than syntactic reasoning (McCoy et al., 2019; Cosma et al., 2024); and in mathematical problem solving, strong in-distribution scores often fail to transfer as problem structure or size shifts (Saxton et al., 2019; Kao et al., 2024; Zhou et al., 2025). This motivates a foundational question:

> *When and why do Transformers learn heuristics over verifiably correct algorithms,*
> *even when the task admits an algorithmic solution?*

To study when Transformers learn algorithms rather than shortcuts, we adopt *graph connectivity* as a controlled testbed. Connectivity offers a unique ground-truth solution: given an adjacency matrix $A$ with self-loops, reachability equals the transitive closure and is computable by classical dynamic programming (Warshall, 1962; Floyd, 1962), so the target is unambiguous. Connectivity is a fundamental algorithmic problem and is one of the most well-studied problems in terms of its complexity (Wigderson, 1992). It is a natural starting point since it sits low in the complexity hierarchy, undirected $s$–$t$ connectivity is known to lie in deterministic logspace L (Reingold, 2008). Recent theory further shows that Transformers with depth growing logarithmically in input length can implement nontrivial parallel algorithms—including connectivity—via repeated squaring-style constructions (Merrill & Sabharwal, 2025). At the same time, connectivity admits simple heuristics based on global statistics such as node degrees or local density, which can be predictive on many graphs but are misleading on others. By combining an unambiguous algorithmic target, a complexity landscape tied to Transformer depth, and natural shortcut baselines, connectivity is a principled stress test of whether training yields multi-step algorithmic composition or superficial cues.

**Our contributions.** Despite theoretical expressivity guarantees, whether gradient descent can help Transformers find the algorithmic solution remains unknown. In this work, our preliminary experiments reveal a clear algorithm-heuristic tension on graph connectivity (see §3.3 and Figure 1):

Transformer models achieve perfect in-distribution accuracy but fail to generalize. We consider a simplified Transformer architecture, the *disentangled Transformer* (Friedman et al., 2023; Nichani et al., 2024) to analyze when training recovers an algorithmic solution rather than a shortcut. Finally we empirically show that the same pattern transfers to standard Transformer models. We summarize our contributions below:

1. *Non-asymptotic capacity tied to diameter.* Prior work establishes log-depth expressivity for connectivity: Transformer models require a depth of $L \in \Theta(\log(n))$ in the number of nodes $n$ (Merrill & Sabharwal, 2025). In Theorem 4.4, we prove a non-asymptotic capacity theorem that depends on *instance difficulty*, characterized by the graph diameter rather than only on the number of nodes $n$. Let $\mathrm{diam}(G)$ denote the maximum shortest-path distance between any two connected nodes. We show that an $L$-layer model solves connectivity on all graphs with $\mathrm{diam}(G) \leq 3^L$. We refer to $3^L$ as the model's *capacity*. We complement this by proving a matching capacity bound of $3^L$, and we empirically validate the diameter-depth scaling by training both disentangled and standard Transformer models.

2. *An algorithm-heuristic decomposition.* We prove in Theorem 4.6 and §4.3 that if the model has certain symmetries such as being invariant to relabellings of the vertices of the graph, the learned weights for a disentangled Transformer are a superposition of an algorithmic and a heuristic channel. We empirically validate that trained models have this invariance property. The algorithmic channel is responsible for multi-hop composition or the computation of matrix powers of the adjacency matrix (via repeated squaring). The heuristic channel determines if two nodes are connected based on the degrees of the two nodes, and similar higher-order generalizations of the degree based on the local neighborhood of the two nodes.

3. *Training dynamics.* Our analysis of the training dynamics reveals a sharp dichotomy driven by the data distribution. For graphs within the model's capacity (diameter $\leq 3^L$), population gradients suppress the heuristic channel and favor the algorithmic channel that implements matrix powering (Theorem C.5). Conversely, when the distribution contains a significant share of beyond-capacity graphs (diameter $> 3^L$) the gradients instead strengthen the heuristic channel, promoting the simple degree-counting shortcut (Theorem C.9). This precise characterization hinges on our exact $3^L$ capacity bound; an asymptotic one, such as the $\mathcal{O}(\exp(L))$ result from Merrill & Sabharwal (2025), would not yield such clear predictive implications.

4. *The Data Lever.* These theoretical insights point to a direct mitigation strategy we call the *data lever*: restricting the training data exclusively to within-capacity graphs. Our experiments in §5 confirm the effectiveness of this approach, showing that it boosts the algorithmic component and improves out-of-distribution robustness (Figure 4), and that these benefits transfer successfully to standard Transformer models (Figure 6).

With graph connectivity as a testbed, our results together pinpoint precise breaking points of Transformers and how the training data influences the learning of generalizable algorithmic components versus brittle heuristics. Our analysis yields a strategy to reduce dependence on heuristics, via the data lever, demonstrating that the theory also has some prescriptive power.

## 2 RELATED WORK

**Computational Complexity and Expressivity of Transformers.** Theoretical analyses aim to define what Transformers can and cannot compute. Although Transformers are universal approximators for continuous sequence-to-sequence functions (Yun et al., 2020), they also face sharp complexity-theoretic limits. Fixed-depth attention struggles with periodic or hierarchical patterns (Hahn, 2020), and standard Transformers are restricted to the complexity class $\mathsf{TC}^0$ (Merrill & Sabharwal, 2023), with hard-attention variants also confined to low-level circuit classes (Hao et al., 2022; Barceló et al., 2024). This computational power can be expanded. Allowing model depth to scale logarithmically with input length enables recognition of regular languages and solves graph connectivity (Merrill & Sabharwal, 2025), while chain-of-thought generation also strictly increases expressivity (Merrill & Sabharwal, 2024). Programmatic abstractions like RASP offer another lens, identifying which algorithms can be implemented in a length-generalizing way (Weiss et al., 2021; Zhou et al., 2023). Empirically for the graph connectivity problem, Fu et al. (2024b) shows frontier LLMs can reach almost perfect performance on small graphs and Saparov et al. (2025) shows transformer has greater difficulty in learning the task when graph size increases. Additionally, Sanford et al. (2024) proves that logarithmic depth is both necessary and sufficient for parallelizable graph tasks, with supporting GraphQA evidence.

**Mechanistic Interpretability of Transformers.** A growing body of work reverse-engineers the *algorithmic circuits* that Transformers learn for tasks like copying, induction, and reasoning (Elhage et al., 2021; Olsson et al., 2022; Wang et al., 2022; Brinkmann et al., 2024). These can range from Fourier-style circuits for modular addition (Nanda et al., 2023; Zhou et al., 2024a) to Newton-like updates for in-context linear regression (Fu et al., 2024a). Researchers validate hypotheses by compiling programs into model weights (Lindner et al., 2023), decompiling models into code (Friedman et al., 2023), and using causal interventions to localize function (Chan et al., 2022; Meng et al., 2022; Yao et al., 2024; Chang et al., 2024). On the other hand, Wen et al. (2023) show that head-by-head mechanistic explanations can mislead: even on a simple parentheses task, transformers spread a stack-like computation across many parts of the network. Finally, theoretical work on inductive biases, like a preference for low-sensitivity functions, helps explain why models often favor robust heuristics over exact algorithms (Vasudeva et al., 2025).

## 3 PROBLEM SETUP AND PRELIMINARY STUDY

### 3.1 GRAPH CONNECTIVITY TASK

**Definition 3.1** (Self-loop-augmented adjacency matrix). *Let $G = (V, E)$ be a finite simple graph with $n$ vertices. We define the **self-loop-augmented adjacency matrix** $A \in \{0, 1\}^{n \times n}$ as: $A_{i,j} = 1$ if $\{v_i, v_j\} \in E$ or $i = j$, and 0 otherwise.*

This definition is equivalent to taking the standard adjacency matrix and adding the identity matrix $(A_G + I_n)$. A key consequence is that the $(i, j)$-th entry of the matrix power $A^k$ counts the number of walks of length $k$ from $v_i$ to $v_j$. With self-loops, these walks may stay at the same vertex from one step to the next. Henceforth, "adjacency matrix" will refer to this self-loop-augmented version.

**Definition 3.2** (Connectivity). *For any graph $G = (V, E)$ with $n$ nodes, we define the connectivity matrix $R \in \{0, 1\}^{n \times n}$ as follows: $R_{i,j} = 1$ if there is a path between $v_i$ and $v_j$ and 0 otherwise. In particular, $R_{i,j} = 1$ if and only if $[A^n]_{i,j} > 0$.*

Our learning objective is to learn models $\mathcal{M} : \{0, 1\}^{n \times n} \to \mathbb{R}^{n \times n}$. For a graph $G$ with adjacency matrix $A$ and connectivity matrix $R$, if $\mathcal{M}$ satisfies $[\mathcal{M}(A)]_{i,j} > 0 \Leftrightarrow R_{i,j} = 1$, then we say $\mathcal{M}$ is *perfect* on graph $G$. We train on Erdős-Rényi $\mathsf{ER}(n, p)$ graphs: on $n$ vertices, each edge is present independently with probability $p$.

### 3.2 TRANSFORMER ARCHITECTURES

We first introduce our setups on standard transformer models without causal attention masking.

**Definition 3.3** (Transformers for Graph Connectivity). *Let $A$ be the self-loop-augmented adjacency matrix of $G$. Fix depth $L$ and hidden width $d > n$. Define the linear read-in and read-out maps*

$$\mathsf{ReadIn}(X) := XW_{\mathrm{in}}, \quad W_{\mathrm{in}} \in \mathbb{R}^{n \times d}, \qquad \mathsf{ReadOut}(H) := HW_{\mathrm{out}}^{\top}, \quad W_{\mathrm{out}} \in \mathbb{R}^{n \times d}.$$

*An $L$-layer single-head transformer model for graph connectivity acts as*

$$\mathsf{TF}_{\Theta}^{L}(A) := \mathsf{ReadOut}\Big(\mathsf{Transformers}^{L}\big(\mathsf{ReadIn}(\bar{A})\big)\Big) \in \mathbb{R}^{n \times n}$$

*where $\mathsf{Transformers}^L$ is a standard pre-norm Transformer with self-attention and with no causal attention masks. There is no additional positional encoding since $I_n$ is already added to the input $A$ as the absolute positional encoding. A full definition can be found in Definition A.1.*

### 3.3 PRELIMINARY STUDY

We train 2-layer Transformer models on $\mathsf{ER}(n = 20, p = 0.08)$ graphs and test them on two out-of-distribution datasets: (1) $\mathsf{2Chain}(n = 20, k = 10)$ graphs with $n$ nodes consisting of two isolated chains each with $k$ nodes, and (2) $\mathsf{2Clique}(n = 20, k = 10)$ graphs with $n$ nodes consisting of two isolated $k$-Cliques. We measure the performance of model $\mathcal{M}$ via an exact match accuracy on our graph distribution $\mathcal{G}$, i.e., the fraction of graphs on which $\mathcal{M}$ is perfect. Formally, it's defined as

$$\mathsf{ExactMatchAcc}(\mathcal{M}, \mathcal{G}) = \mathbb{E}_{G=(V,E) \in \mathcal{G}} \left[ \prod_{v_i, v_j \in V} \mathbf{1}\left\{ [\mathcal{M}(A_G)]_{i,j} = [R_G]_{i,j} \right\} \right].$$

**Transformers Fail to Generalize.** As shown in Figure 1, the 2-layer Transformer model is able to achieve almost perfect exact match accuracy on the held-out set of the training distribution. However, it fails to learn an algorithmic solution which would transfer to other distributions. When the

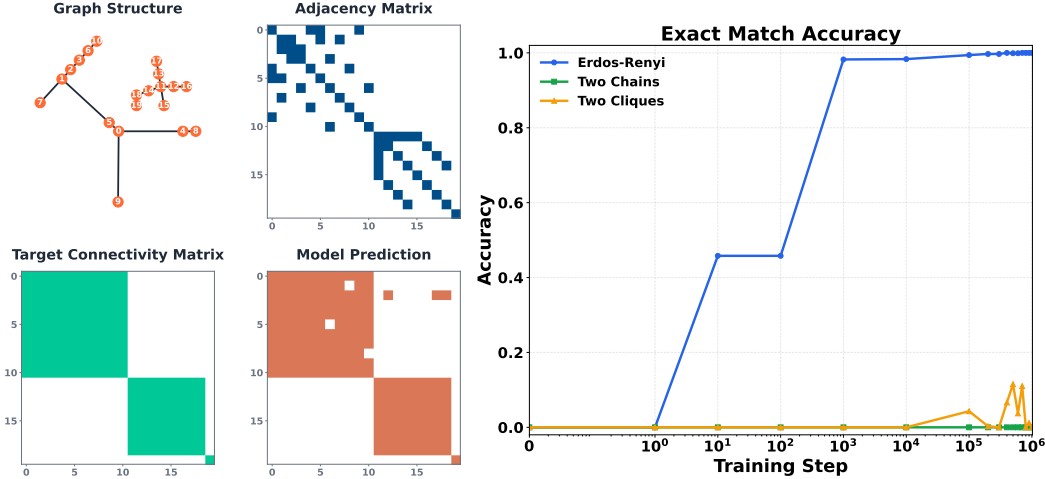

Figure 1: We train 2-layer Transformer models on Erdős-Rényi graphs. (**Left**) Visualization of input, target and model prediction of a sample graph. (**Right**) Although trained models are able to predict perfectly on every edge within distribution, they failed to generalize to out-of-distribution graphs such as graphs with two isolated chains or cliques.

model is tested on the 2Chain and 2Clique distributions, its exact match accuracy falls to nearly zero, indicating over-fitted heuristics have dominated the model prediction. We repeat the experiments via extensive hyperparameter search and scaling up the number of layers, but all models fail to generalize. This motivates us to further investigate this generalization failure on *why* transformers prefer learning heuristics, *when* this happens during the training dynamics, and *if* models can be mitigated to learn actual algorithmic solutions instead of overfitting heuristics.

## 4 THEORY

### 4.1 DISENTANGLED TRANSFORMER

To understand the generalization failure in §3.3 theoretically, we pivot to a simplified ***disentangled Transformer*** which helps us with not only expressivity/capacity analysis in §4.2 but also with training dynamics analysis in §4.3. In the disentangled Transformer, each attention block appends its output as a new coordinate slice of the residual stream rather than summing, so the representation dimension grows with depth and the read/write pathways become traceable (Friedman et al., 2023; Nichani et al., 2024). This model is more than a theoretical convenience; it serves as a valid proxy for its standard counterpart. Nichani et al. (2024) show that any standard attention-only transformer can be re-expressed as a disentangled model by specializing attention to implement feature concatenation, and Chen et al. (2024) adopt this architecture precisely because it preserves the computations of interest while being markedly more amenable to theoretical analysis. We now formalize the model.

**Definition 4.1** (Disentangled Transformer for Graphs). *Let $n$ be the number of nodes for any graph $G$ with adjacency matrix $A \in \mathbb{R}^{n \times n}$. Let $L$ be the depth of the disentangled transformer, and $\{d_0, d_1, \cdots, d_L\}$ be the set of dimensions of its hidden states with $d_\ell = 2^{\ell+1}n$. Let $\{W_\ell\}_{\ell=1}^L$ be the attention matrices with $W_\ell \in \mathbb{R}^{d_{\ell-1} \times d_{\ell-1}}$. Let $W_O \in \mathbb{R}^{n \times d_L} = [I_n, \cdots, I_n]$ be the output matrix. Let $\Theta = \{W_\ell\}_{\ell=1}^L$. An $L$-layer disentangled transformer $\mathsf{TF}_\Theta^L$ acts on any graph's self-loop augmented adjacency matrix $A$ by*

$$\textit{Input hidden state} \qquad h_0 := [I_n, A] \in \mathbb{R}^{n \times d_0}$$

$$\textit{Hidden states at layer } \ell \qquad h_\ell := [h_{\ell-1}, \mathrm{Attn}\,(h_{\ell-1}; W_\ell)] \in \mathbb{R}^{n \times d_\ell} \tag{1}$$

$$\textit{Output layer} \qquad \mathsf{TF}_\Theta^L(A) := h_L W_O^\top$$

$$\textit{where} \qquad \mathrm{Attn}\,(h_{\ell-1}; W_\ell) := \frac{1}{n}\mathsf{ReLU}\left(h_{\ell-1} W_\ell h_{\ell-1}^\top\right) h_{\ell-1} \tag{2}$$

*We remark that $h_\ell \in \mathbb{R}^{n \times d_\ell}$ where $d_\ell = 2^{\ell+1}n$ grows exponentially with respect to $\ell$.*

## 4.2 Expressivity and Capacity

If a 2-layer transformer fails to generalize in §3.3, should we attribute this to the architecture's expressivity? We argue not. Theorem 4.3 shows that an $L$-layer disentangled transformer can implement the matrix powering algorithm and is perfect on graphs of diameter at most $3^L$. Moreover, Theorem 4.4 shows this $3^L$ threshold is tight and exact. To make this precise, we first formalize graph distance and diameter in Definition 4.2.

**Definition 4.2** (Graph distances and diameter). *Let $G = (V, E)$ be a finite, simple, undirected graph. Following standard definitions, for $u, v \in V$, we let $d_G(u, v)$ be the* shortest-path distance *between $u, v$, which is finite if they are connected and infinite otherwise. For a connected component, we define its* diameter *to be the longest path length within the component.*

*Throughout, we define the diameter of a graph, denoted* $\mathrm{diam}(G)$, *to be the maximum diameter among its connected components. Note this differs from the common convention on disconnected graphs, where the latter sets $\max_{u,v} d_G(u, v) = \infty$. Ours is always finite.*

We begin by establishing the expressive power of the disentangled transformer, showing that with sufficient depth, it can implement the correct matrix powering algorithm to solve connectivity.

**Theorem 4.3** (Expressivity). *There exists an $L$-layer disentangled transformer that makes perfect predictions for every graph $G$ satisfying $\mathrm{diam}(G) \le 3^L$.*

*Sketch of proof.* For all $\ell$, setting $W_\ell = I_{d_{\ell-1}}$ suffices. These choices of weights implements the matrix powering algorithm $\sum_{j=0}^{\mathrm{diam}(G)} \alpha_j A^j$ with nonnegative coefficients $\alpha_j > 0$. $\square$

The expressivity result shows what is possible, we next show a capacity bound which reveals the model's inherent limitations. We now prove a tight, non-asymptotic upper bound on the graph diameter an $L$-layer model can handle, linking model depth directly to instance difficulty.

**Theorem 4.4** (Capacity). *Fix $L \ge 1$ and let $\mathsf{TF}_\Theta^L$ be an $L$-layer disentangled transformer on $n = \Omega(3^L)$ nodes. Further assume that the weights $W_\ell \ge 0$ for each $\ell$. Then there exists a graph $G$ with diameter $> 3^L$ on which $\mathsf{TF}_\Theta^L(A)$ is not perfect. In other words, diameter $3^L$ upper bounds the capacity of any $L$-layer disentangled transformer. In particular, taking $n \ge (7/3) \cdot 3^L + 2$ suffices.*

*Sketch of proof.* We split by whether a false positive across different connected components occurs at some intermediate layer; the full proof can be found in Section B.2.

*Case 1 (False positive occurs Lemma B.1).* Suppose for some graph $H$ and a layer $\ell$, a positive score appears at $(u, v)$ in different components. Take $\ell$ minimal (the earliest layer across all graphs where a false positive emerges). We will create a new graph $G$ that (i) preserves this false positive on $(u, v)$ and (ii) contains a path of length $> 3^L$. To do so, we backtrack the computation DAG, tracing the "sources" that contribute to the false positivity of $(u, v)$. This gives us two subgraphs (one for $u$, one for $v$) with roots at layer $\ell - 1$ that we call *certificates*. Because $\ell$ is minimal, every previous layer is free of false positives; hence, the two certificates induce two disjoint sets of graph nodes. Finally, create a new graph $G$ which embeds these two certificates and arranges all other nodes into a long chain. Then $G$ has the properties we seek.

*Case 2 (No false positives Lemma B.2).* Suppose now that no intermediate layer has false positives. We show that "information" spreads no faster than $3^L$ so that it never predicts "Yes" on node pairs with distance beyond $3^L$. We first apply the no-false-positives assumption on the empty (self-loops-only) graph. Inductively, each column of each hidden states is supported on exactly one row, which ranges from 1 to $n$. This naturally gives a "label" for each column in each hidden states. The crux of the proof is to inductively show that at layer $\ell$, two columns can "share" information, thereby creating a positive score on $(u, v)$, only if their labels, interpreted as graph nodes, are within distance $3^\ell$. Consequently, a no-false-positive model cannot recognize a connected pair with distance $> 3^L$.

In both cases one can construct graphs with diameters $> 3^L$ on which $\mathsf{TF}_\Theta^L$ is not perfect. $\square$

Given the tight $3^L$ capacity bounds for transformers, it is natural and crucial to introduce a dichotomy around the $3^L$ capacity. For any connected node pair $(u, v)$, they are said to be within capacity if $d_G(u, v) \le 3^L$ and beyond capacity otherwise. Formally, we define the dichotomy as follow:

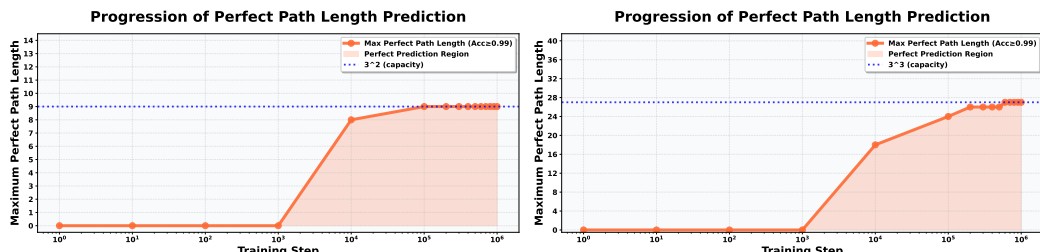

Figure 2: **Capacity of Disentangled Transformers.** We train 2-layer (**left**) and 3-layer (**right**) disentangled transformers on $\mathsf{ER}(n=24)$ and $\mathsf{ER}(n=64)$ graphs respectively. When evaluated on hold-out sets, both models can only make reliable predictions ($\geq 99\%$ accuracy) on node pairs $u, v$ if and only if $d_G(u,v) \leq 3^L$. It resonates with our theoretical observations in Theorem 4.4.

**Definition 4.5** (Within-capacity and beyond-capacity pairs at depth $L$). *Fix a graph $G$ and a depth $L$. We say a pair of nodes $(i,j)$ is **within capacity** if $[A^{3^L}]_{i,j} > 0$ and **beyond capacity** otherwise. In other words, a pair $(i,j)$ is within capacity iff their shortest-path distance is $\leq 3^L$.*

### 4.3 TRAINING DYNAMICS

If a capable 2-layer Transformer is able to perfectly predict connectivity up to path length $3^2 = 9$, and the 2Chain$(n=20, k=10)$ dataset does not contain longer paths, why didn't the Transformer model in §3.3 learn the algorithm? In this section, we show that this is because the training distribution contains too many graphs beyond the $3^L$ capacity, and those samples reward a global shortcut over algorithm. Equipped with Theorem 4.6, we can analyze the gradient dynamics in the two-channel parameterization (a superposition of heuristic and algorithmic channels). Theorems C.5 and C.9 give a simple criterion made possible by the exact $3^L$ characterization: if within-capacity pairs dominate, the algorithmic channel wins; if beyond-capacity pairs prevail, the shortcut wins.

**Parameterizing Model Weights.** To analyze the gradient dynamics, we first identify the relevant parameter space. Our data and targets are symmetric under node relabeling: the ground truth mapping maps $A \mapsto R$ and $PAP^\top \mapsto PRP^\top$ for any permutation $P$. We also empirically observe that models trained from scratch on these graphs rapidly converge to a layerwise equivariant state (Figure 7). Based on these observations, we analyze the dynamics within the subspace of layerwise permutation-equivariant weights. The following Theorem formally defines layerwise equivariance and characterizes exactly what weights look like under this notion.

**Theorem 4.6** (Layerwise Permutation-Equivariant Parameterization). *Suppose an $L$-layer Disentangled Transformer $\mathsf{TF}_\Theta^L$ has non-negative weights, i.e., $W_\ell \geq 0$ for all $\ell$. Let $K_{\ell-1} = 2^\ell$. Then $\mathsf{TF}_\Theta^L$ is layer-wise permutation equivariant, i.e., for each $\ell$ and any hidden states $h \in \mathbb{R}^{n \times d_{\ell-1}}$,*

$$\mathrm{Attn}(Ph(I_{K_{\ell-1}} \otimes P^\top); W_\ell) = P\,\mathrm{Attn}(h; W_\ell)\,(I_{K_{\ell-1}} \otimes P^\top),$$

*if and only if each layer weight $W_\ell$ decomposes as $W_\ell = A_\ell \otimes I_n + B_\ell \otimes J_n$ for some $A_\ell, B_\ell \in \mathbb{R}^{2^\ell \times 2^\ell}$ for all $\ell$, where $\otimes$ denotes the Kronecker product and $J_n = \mathbf{1}\mathbf{1}^T$ the all-ones $(n \times n)$ matrix.*

It immediately follows that this subspace contains a canonical algorithmic solution (e.g. the identity construction $W = I$ used in Theorem 4.3). Furthermore, Theorem C.2 (in Appendix) shows that for *any* capacity-reaching model, the symmetric component of the weights, which drives the attention mechanism, must lie purely in the $I_n$-channel. Finally, algebraically this parameterization is closed under gradients (Theorem C.4), and this allows us to decompose the learning process into the competition between two functionally distinct channels, discussed next.

**The Two-Channel Regime.** Under the conditions of Theorem 4.6, each layer weight splits as $W_\ell = A_\ell \otimes I_n + B_\ell \otimes J_n$. We prove this yields two functionally distinct channels (cf. equation 11):

- ***The algorithmic $I_n$-channel*** $(A_\ell \otimes I_n)$. This channel preserves the locality of the graph structure. It implements *matrix powering* algorithm: across layers, it composes "$k$-hop features" by computing $A^k$, allowing the final readout layer to aggregate $\sum \alpha_j A^j$ in Theorem 4.3.
- ***The heuristic $J_n$-channel*** $(B_\ell \otimes J_n)$. The factor $J_n = \mathbf{1}\mathbf{1}^\top$ is rank-1 and broadcasts information globally. Since $J_n x = (\mathbf{1}^\top x)\mathbf{1}$ and $AJ_n = (A\mathbf{1})\mathbf{1}^\top = \mathbf{d}\,\mathbf{1}^\top$, this channel computes global statis-

tics based on node degrees $d$ and walk counts $A^k \mathbf{1}$. Consequently, the $B_\ell$ parameters contribute to the *degree-counting heuristics* shared across nodes via broadcast.

**Training Dynamics.** Under this algorithmic-herustic dual-channel view, we can track the evolution of the two parameters, $A_\ell$ and $B_\ell$. To rigorously analyze the gradient flow, we adopt the following assumptions regarding the data and the objective.

**Assumption 4.7.** *1. **Data Distribution.** Let $\mathsf{ER}(n, p)$ be the Erdős-Rényi distribution with edge-probability $p \in (0, 1)$. Assume $\mathbb{P}_{G \sim \mathsf{ER}(n,p)}\{G \text{ is disconnected}\}$ is bounded away from 0.*

*2. **Nonnegativity & Equivariant Parameterization.** We assume Theorem 4.6: for each layer $\ell$, assume $W_\ell \geq 0$ and $W_\ell$ can be decomposed as follows:*

$$W_\ell \;=\; A_\ell \otimes I_n \;+\; B_\ell \otimes J_n, \qquad A_\ell, B_\ell \in \mathbb{R}^{2^\ell \times 2^\ell}. \tag{3}$$

*3. **Surrogate Loss.** Given scores $Z := \mathsf{TF}_\Theta^L(\cdot) \in \mathbb{R}_{\geq 0}^{n \times n}$, define the link $\phi(z) := 1 - e^{-\alpha z}$ with $\alpha > 0$,[1] the entrywise Bernoulli cross-entropy with respect to the connectivity matrix $R$ is*

$$\mathcal{L}(Z; R) \;:=\; -\sum_{i,j} \big( R_{i,j} \log \phi(Z_{i,j}) + (1 - R_{i,j}) \log(1 - \phi(Z_{i,j})) \big). \tag{4}$$

*Its gradient with respect to $Z$ is $\frac{\partial \mathcal{L}}{\partial Z} \;=\; \alpha \, (1 - R/\phi(Z)) \;\in\; \mathbb{R}^{n \times n}$.*

Our analysis reveals that the training process consists of two distinct phases.

**Phase 1: Both channels pick up easy examples**. In early updates, both channels quickly ramp up mass because there are plenty of within-component, within-capacity pairs. Concretely, the local $I$-channel composes neighborhood information, while the global $J$-channel can also boost under-predicted positives without facing much penalty (Remark C.7). Phase 1 i s transient and ends once those easily connected pairs are mostly saturated. In Figure 3 (left), it only occupies around $2 \cdot 10^2$ steps out of $10^4$ total.

**Phase 2: Data determines which channel wins**. Once in this regime, the growth of $B_\ell$ is determined by the population-level balance (see Theorem C.5 for full details and notations): informally,

$$\text{Derivative of } B_\ell \;\propto\; \mathbb{E}\left[ \underbrace{\sum_{R_{i,j}=0} D_{i,j}}_{\substack{\text{penalty from} \\ \text{cross component}}} - \underbrace{\sum_{R_{i,j}=1} \frac{1 - \phi(Z_{i,j})}{\phi(Z_{i,j})} D_{i,j}}_{\substack{\text{weighted reward on} \\ \text{under-predicted positives}}} \right]. \tag{5}$$

There are two outcomes. If batches carry a significant mass of disconnected, within-capacity graphs, the cross-component penalty dominates (derivative $> 0$). This suppresses the heuristic $J$-channel ($B_\ell \to 0$), leaving only the algorithmic $I$-channel (Theorem C.5(ii)). As seen in Figure 4 (right, restricted), the $A$-share climbs toward 1 while the $B$-share decays to 0. Conversely, if the distribution contains a significant share of connected, beyond-capacity graphs, the reward term dominates (derivative $< 0$), so gradients strengthen the $J$-channel. This promotes the simple degree-counting shortcut over the algorithm.

A helpful way to interpret Phase 2 is via a mixture of graph types. The per-sample results (Theorem C.9) say: within-capacity graphs reward the local $I$-channel and, when disconnected, penalize any (falsely) activated $J$-channel; large-diameter connected graphs do the opposite and push $J$-channel up. Aggregating over the data distribution leads to the conclusion that the outcome of Phase 2 depends on the fraction of beyond-capacity connected graphs (Remark C.13). Figure 5 visualizes the effects of various mixtures.

## 5 EXPERIMENTS

We test the theory in two parts. First, in §5.1 we verify the $3^L$ capacity threshold (Theorems 4.3 and 4.4) by measuring the maximum reliable path length one model can handle perfectly. Then, we trace training dynamics by projecting learned weights into the algorithmic $A \otimes I$ channel and

---

[1]It is possible that $R_{i,j} = 1$ while $Z_{i,j} = 0$, resulting in undefined gradient $\partial \mathcal{L}/\partial Z$. To circumvent this, we approximate via $\phi_\epsilon = 1 - (1 - \epsilon)e^{-\alpha z}$. All subsequent analyses hold verbatim by replacing $\phi$ with $\phi_\epsilon$.

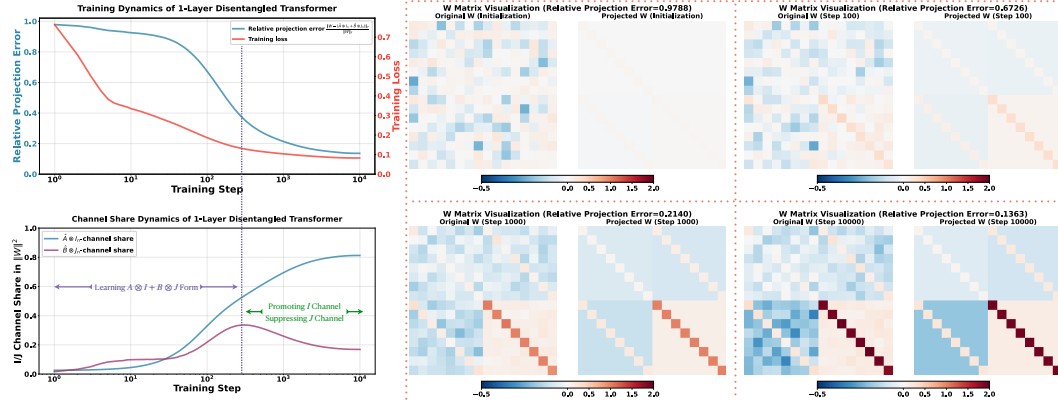

Figure 3: **Training Dynamics of Disentangled Transformers.** We train a 1-layer disentangled transformer on graphs from $\mathsf{ER}(n = 8, p = 0.2)$ distribution. Weight $W$ will approximately approach to $A \otimes I_n + B \otimes J_n$ form. (**Left**) There are two major phases during training, where during Phase 1, model focuses on learning the equivariant parameterizations so both $I$ and $J$ channel's share of energy in $W$ increases, and during Phase 2, the algorithmic $I$-channel is promoted and the heuristic $J$-channel is suppressed. (**Right**) Visualization of the learned weights during training and its projection to the closest $\hat{W} = \hat{A} \otimes I_n + \hat{B} \otimes J_n$ form.

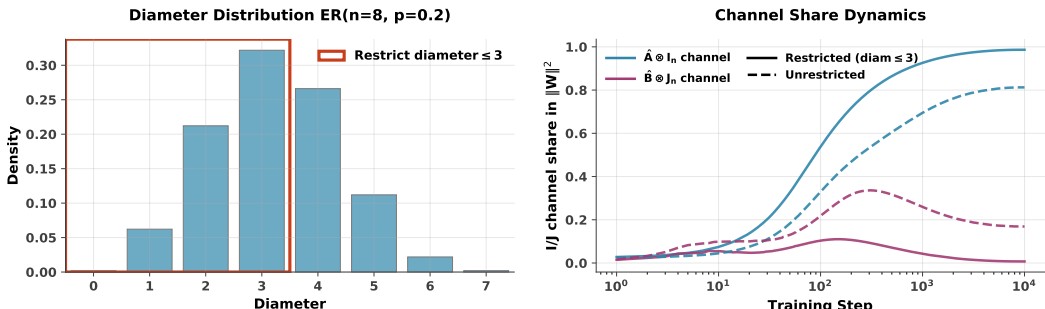

Figure 4: Following insights from Theorems C.5 and C.9, we repeat the same experiment setup as in Figure 3 but only training on within-capacity graphs (see Definition 4.5 and the **left** figure). As shown in the **solid** lines in the **right** figure, restricting training samples by capacity pushes the energy share of the **algorithmic** mechanism (the $A \otimes I_n$ channel) further to nearly 100% in the weight $W$. It simultaneously prevents the growth of the **heuristic** portion (the $B \otimes J_n$ channel).

the heuristic $B \otimes J$ channel (Theorem 4.6). Next, in §5.2 we introduce a simple data lever that upweights within-capacity graphs, and shows this simple method suppresses the heuristic and promotes the algorithmic channel, as predicted by Theorems C.5 and C.9. Finally, we show this data lever prescribed by our theoretical analysis on disentangled transformers can transfer back to standard Transformers, and boost their generalization capabilities.

## 5.1 CAPACITY AND TRAINING DYNAMICS

$L$**-layer Transformers Hit Their Capacity at Exactly** $3^L$**.** We train disentangled transformers with 2 layers or 3 layers on Erdős-Rényi graphs with 24 or 64 nodes respectively. As shown in Figure 2, neither of the two models could make reliable predictions on node pairs $(u, v)$ with $d_G(u, v) > 3^L$ but their predictions on node pair with $d_G(u, v) \le 3^L$ are almost perfect with an $> 99\%$ accuracy. As shown in Figure 9, a 2-layer standard Transformer model also has the same capacity. It resonates with our exact capacity bound of disentangled transformers in Theorem 4.4 and justifies our dichotomy in Definition 4.5. Overall, for any graph $G = (V, E)$, the decisive factor for transformer model depth is not simply the asymptotic $\Theta(\log |V|)$ relation to the number of nodes $n = |V|$ but more importantly the non-asymptotically exact relation to $\log_3 \mathrm{diam}(G)$. It allows us to better understand the training dynamics in §4.3 with a dichotomy, something impossible without an ex-

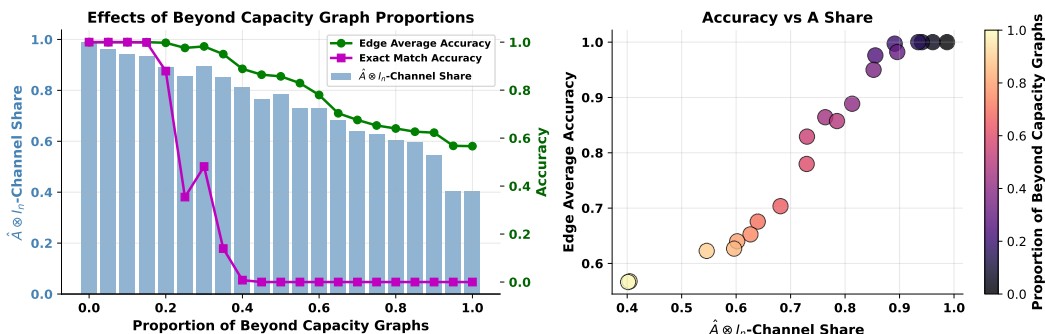

Figure 5: We vary the proportion of beyond-capacity graphs, and train the same disentangled transformer on stratified ER distribution and test on the same OOD 2Chain distribution. We find that Transformers are robust towards a small amount of noises (beyond-capacity graphs). Although the $W$ is not exactly in the $A \otimes I_n$ form, the model still performs perfectly when the energy share of $I$-channel dominates (beyond roughly 90%). As shown on the right, there is a linear correspondence between model accuracy (averaged over individual node pairs) and algorithmic channel share.

act relation. We note that our experiments did not enforce non-negativity of model weights and it demonstrates the predictive power of our theoretical analysis.

**Transformers Learn an Algorithm-Heuristic Mixture.** In understanding the training dynamics, we first train a 1-layer disentangled transformer model on $\mathsf{ER}(n = 8, p = 0.2)$ graphs. We did not enforce any parameterization assumptions here. As shown in Figure 3, a randomly initialized $W$ can converge to approximately a $A \otimes I_n + B \otimes J_n$ decomposition for some matrices $A, B \in \mathbb{R}^{2 \times 2}$ and $J_n = \mathbf{1}\mathbf{1}^\top \in \mathbb{R}^{n \times n}$. Such parameterization also applies to deeper models as shown in Figure 8. They show the applicability of our decomposition Theorem 4.6. Then, we project the final weight $W$ onto this algebra as $\hat{W} = \hat{A} \otimes I_n + \hat{B} \otimes J_n$ by minimizing $\|W - \hat{W}\|_F$ and observe that the energy share (see §D) of $\hat{A} \otimes I_n$ in $\|W\|_F^2$ increases as training progresses but the share of $\hat{B} \otimes J_n$ first increases and then decreases, showing interesting training dynamics to be studied.

### 5.2 ENCOURAGING TRANSFORMERS TO LEARN ALGORITHMS OVER HEURISTICS

Now that we understand *why* Transformers and disentangled transformer models learn heuristics that hurt their algorithmic computations (as shown in Figures 1 and 3), a natural question is whether we can mitigate it and encourage the models to up-weight the algorithm channel.

**Mitigation via the Data Lever.** We propose a data-centric method: instead of training on all graphs from the ER distribution, we up-weight graphs whose $\mathrm{diam}(G)$ is within capacity following the dichotomy in Definition 4.5. We dissect $\mathcal{G} = \mathcal{G}_\leq \sqcup \mathcal{G}_>$ into two sub-distributions where $\mathcal{G}_\leq = \{G \in \mathcal{G} : \mathrm{diam}(G) \leq 3^L\}$ only includes graphs containing no beyond capacity node pairs and $\mathcal{G}_>$ includes the rest. In Figure 4, we only train the 1-layer disentangled transformer on $\mathsf{ER}_\leq$, and then find the algorithmic $\hat{A} \otimes I_n$ channel is significantly promoted so that the learned weight only contains the algorithm channel. Furthermore, we find at-capacity graphs are crucial. In the case of Figures 10 and 11, where no graphs are to have $\mathrm{diam}(G) > 2$, model also fails to learn generalizable solutions due to transformers' poor length generalization abilities. It implies simply scaling up the model depth won't equip it algorithmic capabilities naturally.

**Robustness to Noise.** To test the predictiveness of our theory, we evaluate if one beyond-capacity node pair is enough to encourage the model learning a heuristic-dominated method. We define $\rho(\mathcal{G}) = \mathbb{E}_{G \in \mathcal{G}} |\{(u, v) \in V, d_G(u, v) > 3^L\}|/n^2$ be the fraction of beyond-capacity node pairs in a graph distribution $\mathcal{G}$. In practice, $\rho$ can be controlled via stratified sampling from the mixture distribution $\mathcal{G}_q = q\mathcal{G}_\leq + (1 - q)\mathcal{G}_>$. In Figure 5, we did stratified sampling between $\mathsf{ER}_\leq$ and $\mathsf{ER}_>$ and find that with a small $\rho(\mathcal{G})$, the model is still able to maintain high energy in the algorithm channel, and make perfect predictions on out-of-distribution 2Chain graphs. It suggests that there exists a small $\rho^\star > 0$ such that the model can still rely on the algorithm-channel to make predictions if the training distribution satisfies $\rho(\mathcal{G}) \leq \rho^\star$.

**Transferability to Standard Transformers models.** Our theory from §4.3 makes a prescriptive suggestion to remove beyond capacity graphs to reduce dependence on heuristics, and Figure 4 demonstrated the effectiveness of this approach on disentangled transformers. We now evaluate this on standard Transformers. We train the same 2-layer Transformers model as in our preliminary study in §3.3 but this time, we train on the restricted distribution $\mathsf{ER}_<$ instead where all graphs $G \in \mathsf{ER}_<$ have $\mathrm{diam}(\widetilde{G}) \leq 3^2 = 9$. As shown in Figure 6, when tested on the OOD 2Chain dataset with maximum chain length 10, the one trained on $\mathsf{ER}_<$ can successfully generalize but the one trained on unconstrained distribution ER cannot.

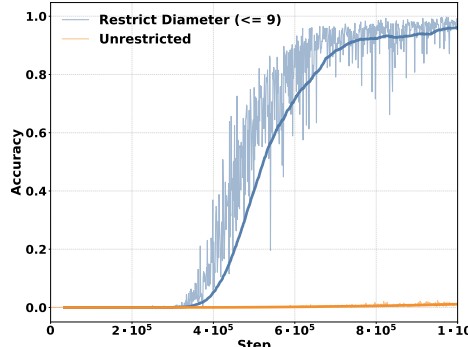

Figure 6: Standard transformer models learn generalizable solutions from within capacity data.

## 6 DISCUSSION AND CONCLUSION

In this paper, we separate expressivity from capacity and training dynamics for Transformers on graph connectivity. We prove that an $L$-layer model can implement matrix powering and is perfect on graphs with $\mathrm{diam}(G) \leq 3^L$, and we show this $3^L$ threshold is tight. The failures in §3.3 are explained by a capacity mismatch: training mass beyond $3^L$ steers learning toward a global shortcut rather than the intended multi-hop algorithmic computation. Our two-channel view makes this explicit and turns generalization into a property of the data distribution: when within-capacity pairs dominate, the algorithmic channel is selected. Experiments confirm the threshold and show that a simple capacity-aware data lever that up-weights within-capacity graphs suppresses the shortcut, promotes out-of-distribution generalization, and transfers to standard Transformers. By pinpointing when a model reaches for a shortcut and showing how simple data choices can steer it towards the true algorithmic solution, we outline a path to systematically control training data and model capacity to enable Transformers to learn solutions that generalize better.

## THE USE OF LARGE LANGUAGE MODELS (LLMs)

LLMs were used only to polish language, such as grammar and wording. These models did not contribute to idea creation or writing, and the authors take full responsibility for this paper's content.

## REPRODUCIBILITY STATEMENT

We have taken several steps to facilitate reproducibility. All theoretical claims (expressivity and the exact $3^L$ capacity; two-channel training dynamics) include formal assumptions, definitions, and proofs (Definition 4.2, Theorems 4.3, 4.4 and 4.6, Assumption 4.7, and Appendices §B and §C). Our experiments use only synthetic data. We include source codes in the supplementary materials.

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

## A  ADDITIONAL DETAILS ON PROBLEM SETUPS AND PRELIMINARY STUDIES

**Definition A.1** (Transformer for Graph Connectivity: full specification)**.**

***Input and output****. Given a simple graph on $n$ nodes with adjacency matrix $A \in \{0,1\}^{n \times n}$, let $\bar{A} = A + I_n$ be its self-loop augmented adjacency matrix. We treat $\bar{A}$ as input embedding: row $i$ is the token for node $i$; column $j$ indexes a feature tied to node $j$. The model outputs an $n \times n$ score matrix $\mathrm{TF}_\Theta^L(A)$, the predicted connectivity matrix.*

***Dimensions and parameters****. Fix depth $L$, hidden dimension $d > n$, number of heads $H$ with $d = H d_h$, and feed-forward width $d_{\mathrm{ff}}$. The parameters we need include:*

$$W_{\mathrm{in}}, W_{\mathrm{out}} \in \mathbb{R}^{n \times d}, \qquad W_{\ell,h}^Q, W_{\ell,h}^K, W_{\ell,h}^V \in \mathbb{R}^{d \times d_h}, \qquad W_\ell^O \in \mathbb{R}^{H d_h \times d}$$

$$W_\ell^{(1)} \in \mathbb{R}^{d \times d_{\mathrm{ff}}}, \ b_\ell^{(1)} \in \mathbb{R}^{d_{\mathrm{ff}}}, \qquad W_\ell^{(2)} \in \mathbb{R}^{d_{\mathrm{ff}} \times d}, \ b_\ell^{(2)} \in \mathbb{R}^d$$

*for $\ell = 1, \ldots, L$ and heads $h = 1, \ldots, H$. We use pre-norm residual blocks with LayerNorm (LN) and GeLU activations. We do not use attention masks or any extra positional encoding; the identity in $\bar{A}$ already pins each token to a node.*

***The forward map****. The read-in is linear: $h^{(0)} = \bar{A} W_{\mathrm{in}} \in \mathbb{R}^{n \times d}$. From there, for each $\ell = 1, \ldots, L$, let $\tilde{h} = \mathrm{LN}(h^{(\ell-1)})$ as we use pre-norm. Within each block:*

| | |
|---|---|
| ***Multi-head self-attention*** | $Q_{\ell,h} = \tilde{h} W_{\ell,h}^Q, \quad K_{\ell,h} = \tilde{h} W_{\ell,h}^K, \quad V_{\ell,h} = \tilde{h} W_{\ell,h}^V$ |
| ***Attention scores*** | $\alpha_{\ell,h} = \dfrac{1}{n} \mathrm{ReLU}(1/\sqrt{d_h} \cdot Q_{\ell,h} K_{\ell,h}^\top), \quad z_{\ell,h} = \alpha_{\ell,h} V_{\ell,h}$ |
| ***Concatenation & residual*** | $z_\ell = [z_{\ell,1} \mid \ldots \mid z_{\ell,H}] W_\ell^O \in \mathbb{R}^{n \times d}, \qquad u_\ell = h^{(\ell-1)} + z_\ell$ |
| ***Feed-forward*** | $\hat{u}_\ell = \mathrm{LN}(u_\ell), \qquad \mathrm{FFN}_\ell(\hat{u}_\ell) = \mathrm{GeLU}(\hat{u}_\ell W_\ell^{(1)} + b_\ell^{(1)}) W_\ell^{(2)} + b_\ell^{(2)}$ |

*and finally $h^{(\ell)} = u_\ell + \mathrm{FFN}_\ell(\hat{u}_\ell) \in \mathbb{R}^{n \times d}$. The read-out is linear: $\mathrm{TF}_\Theta^L(A) = h^{(L)} W_{\mathrm{out}}^\top \in \mathbb{R}^{n \times n}$.*

**Metrics for Permutation Equivariance.** Let $P \in \mathcal{S}_n$ be the corresponding permutation matrix for any $\sigma \in \mathcal{S}_n$. For a given graph adjacency matrix $A$, we compute the model's prediction in respect to $P_\sigma$ as $\mathcal{M}(P_\sigma A P_\sigma^\top)$. Now we define a equivariance consistency metric, **Equivariance Consistency via Frobenius Cosine Similarity**:

$$\mathrm{ConsFrob}(\mathcal{M}) = \mathbb{E}_{\sigma \in \mathcal{S}_n, A \in \mathcal{G}} \left[ \frac{\langle \mathcal{M}(P_\sigma A P_\sigma^\top), P_\sigma \mathcal{M}(A) P_\sigma^\top \rangle_F}{\|\mathcal{M}(P_\sigma A P_\sigma^\top)\|_F \|P_\sigma \mathcal{M}(A) P_\sigma^\top\|_F} \right] \tag{6}$$

When measuring intermediate model computations, this metric is modified depending on the model type. For standard transformer models, $\mathcal{M}^\ell$ computes the final Readout to the hidden states at layer $\ell$. For Disentangled transformers we are computing $P_\sigma \mathcal{M}^\ell(A)(P_\sigma \otimes I_n)^\top$.

# B  DETAILS FOR CAPACITY

## B.1  EXPRESSIVITY

**Theorem 4.3.** There exists an $L$-layer disentangled transformer that makes perfect predictions for every graph $G$ satisfying $\mathrm{diam}(G) \leq 3^L$.

*Proof.* Set $W_\ell = I_{d_{\ell-1}}$ for all layers and note that all matrices are entrywise nonnegative, so ReLU and the factor $1/n$ never changes supports. With $h_0 = [I \mid A] = [A^0 \mid A^1]$ and update $h_\ell = [h_{\ell-1} \mid (h_{\ell-1}h_{\ell-1}^\top)h_{\ell-1}/n]$, we can show by induction that every $n \times n$ block of $h_\ell$ lies in $\mathrm{span}\{A^0, \ldots, A^{3^\ell}\}$, and that some block contains $A^{3^\ell}$ with a positive coefficient. Indeed, the base case holds trivially; for the inductive step, if a block within $h_{\ell-1}$ contains $A^m$, then $(h_{\ell-1}h_{\ell-1}^\top)h_{\ell-1}$ contains $A^{2m}A^m = A^{3m}$. Finally, the readout simply sums over all these blocks, so $\mathrm{supp}(\mathrm{TF}_\Theta^L(A)) = \mathrm{supp}(A^{3^L})$.

Finally, because $A$ has self-loops, supports are monotone in power and stabilizes at $t \geq \mathrm{diam}(G)$. Thus, if $\mathrm{diam}(G) \leq 3^L$ we get $\mathrm{supp}(\mathrm{TF}_\Theta^L(A)) = \mathrm{supp}(A^{\mathrm{diam}(G)})$. $\qquad\square$

## B.2  CAPACITY

**Theorem 4.4.** Fix $L \geq 1$ and let $\mathrm{TF}_\Theta^L$ be an $L$-layer disentangled transformer on $n = \Omega(3^L)$ nodes. Further assume that the weights $W_\ell \geq 0$ for each $\ell$. Then there exists a graph $G$ with diameter $> 3^L$ on which $\mathrm{TF}_\Theta^L(A)$ is not perfect. In other words, diameter $3^L$ upper bounds the capacity of any $L$-layer disentangled transformer. In particular, taking $n \geq (7/3) \cdot 3^L + 2$ suffices.

For each layer $\ell$ we define the post-ReLU score $R_\ell = \mathrm{ReLU}(h_{\ell-1}W_\ell h_{\ell-1}^\top)$. The proof of the theorem will be partitioned into two branches: whether some intermediate $R_\ell$ gives a false positive on some graph, or all $R_\ell$'s are free of false positives on all graphs. We say a pair of nodes $(u, v)$ from $G$ is a *witness* to false positives if they belong to different connected components while $R_\ell(G)_{u,v} > 0$. Throughout this section, we set $n \geq (7/3) \cdot 3^L + 2$.

**Lemma B.1.** *Assume the setup in Theorem 4.4. Suppose there exist some $n$-node graph, a layer index $\ell^* \in \{1, \ldots, L\}$, and vertices $u, v$ belonging to different connected components of the graph such that $(R_{\ell^*})_{u,v} > 0$. Further assume that $\ell^*$ is globally minimal, in the sense that for all $n$-node graphs and all $\ell < \ell^*$, the corresponding $R_\ell$ has no false positive entries across components. Then there exists a graph $G$ such that $\mathrm{diam}(G) > 3^L$, $\mathrm{TF}_\Theta^L(A(G))_{u,v} > 0$, where $u, v$ lie in different connected components of $G$.*

*Proof.* The proof roughly partitions into two parts. In the first half, we backtrack the computation DAG, tracing the "sources" that contribute to the false positiveness of $(u, v)$. This gives us subgraphs which we call *certificates* that, if kept untouched, suffice to guarantee a false positiveness of $(u, v)$. In the second half, we construct a graph $G$ that preserves these certificates while also containing a path of length $> 3^L$ disjoint from both certificates, and we show that $\mathrm{TF}_\Theta^L$ preserves false positiveness of $(u, v)$ on $G$, thereby proving the claim.

STEP 1. CONSTRUCTING THE CERTIFICATES. Intuitively, since $R_{\ell^*}(H)_{u,v} > 0$, there exist column indices $p, q$ with $(W_{\ell^*})_{p,q} > 0$, $h_{\ell^*-1}(H)_{u,p} > 0$, and $h_{\ell^*-1}(H)_{v,q} > 0$. We will backtrack the entries that contribute to the positiveness of the hidden states entries $h_{\ell^*-1}(H)_{u,p}$ and $h_{\ell^*-1}(H)_{v,q} > 0$ in the computation DAG, iteratively visiting previous layers. Formally, we define a *certificate* for an entry $h_t(H)_{i,c} > 0$ to be a small tree whose nodes are triples [of form (layer, row, column)] recording earlier entries that must be positive to guarantee that the current one is positive. The root is $(t, i, c)$ and we build it top-down by repeating one of the two rules until we hit the first layer, which we know looks like $[I_n \mid A]$. We now describe how to backtrack. Since $h_t = [h_{t-1} \mid \mathrm{Attn}(h_{t-1}; W_t)]$, we split the recursion on layer $t$ into two cases: whether the entry lies in the first half ($h_{t-1}$) or the second half ($\mathrm{Attn}(h_{t-1}; W_t)$).

- (First half) If column $c$ is in the inherited block of $h_t$, add a single child $(t-1, i, c)$ to $(t, i, c)$, as the value is simply copied from the previous layer $h_{t-1}$.

- (Second half) If column $c$ is in the newly appended block, then by definition

$$h_t(H)_{i,c} = \frac{1}{n} \sum_k R_t(H)_{i,k} h_{t-1}(H)_{k,c'} \qquad \text{for some } c',$$

and since this is a sum of nonnegative terms, there exists at least one $k$ with $R_t(H)_{i,k} > 0$ and $h_{t-1}(H)_{k,c'} > 0$. In turn,

$$R_t(H)_{i,k} = \sum_{r,s} h_{t-1}(H)_{i,r}(W_t)_{r,s} h_{t-1}(H)_{k,s} > 0,$$

which implies that there exist indices $r, s$ with $(W_t)_{r,s} > 0, h_{t-1}(H)_{i,r} > 0$, and $h_{t-1}(H)_{k,s} > 0$. Thus, for such $(t, i, c)$, we create three children:

$$(t-1, k, c'), \qquad (t-1, i, r), \qquad (t-1, k, s).$$

Now let $s(t)$ denote the maximal number of vertices needed to realize a single certificate for some entry $h_t(\cdot) > 0$ by the recursive procedure above. At $t = 0$ we may assume $s(0) \leq 2$. The recursion gives $s(t) \leq 3s(t-1)$, so $s(t) \leq 2 \cdot 3^t$. Since $u, v$ lie in different connected components of $H$, and $\ell^*$ is minimal, every index $k$ selected by a certificate at any layer $t \leq \ell^* - 1$ stays within the same component as its $i$, so the two certificates induce trees $T_u, T_v$ that occupy disjoint vertex sets $S_u, S_v$, with $|S_u \cup S_v| \leq 4 \cdot 3^{\ell^*-1} \leq 4 \cdot 3^{L-1}$ vertices.

STEP 2. BUILDING A NEW GRAPH. Initialize $G$ to the edgeless graph, keeping node isolated. We then embed $T_u, T_v$ onto $G$ by adding edges according to the trees. Finally, we connect every vertex outside $S_u \cup S_v$ arbitrarily into a long chain.

We claim $G$ is the graph we seek. On one hand, every sum used by the certificates is a sum of nonnegative terms, and we have preserved a strictly positive summand at each step that appears in the tree. Hence $R_{\ell^*}(G)_{u,v} > 0$ with $u, v$ also disconnected in $G$. On the other hand, under the choice of $n$ specified by Theorem 4.4, there exist at least $3^L + 2$ vertices outside $S_u \cup S_v$, so connecting them into a long path guarantees $\operatorname{diam}(G) > 3^L$. The claim then follows. $\qquad \square$

**Lemma B.2.** *Assume the setup in Theorem 4.4. Further assume that for every $n$-node graph $G$ and every layer $\ell \in \{1, \ldots, L\}$, the post-ReLU scores $R_\ell(G)$ has no positive entry between distinct connected components of $G$. Then, for every graph $G$ and every $u, v \in V(G)$, if $\mathsf{TF}_\Theta^L(A(G))_{u,v} > 0$, we must have $\operatorname{dist}_G(u, v) \leq 3^L$. Consequently, if $G$ contains a connected component of diameter $> 3^L$ then $\mathsf{TF}_\Theta^L$ is not perfect on $G$.*

*Proof.* Under the no-false-positives assumption, the idea is to show that "information" spreads no faster than power base 3 so $\mathsf{TF}_\Theta^L$ never predicts "Yes" on node pairs with distance beyond $3^L$. Concretely, columns exchange information as attention scores are calculated. We first define the "distances" between columns by giving each column a label $\in \{1, \ldots, n\}$, and then show that by layer $\ell$, two columns can "share" information if and only if their labels, *interpreted as graph nodes, are within distance $3^\ell$.*

STEP 1. GIVING EACH COLUMN A LABEL. We first consider trivial graph $G_0$ with $n$ isolated nodes: immediately $h_0(G_0) = [I_n \mid I_n]$ and, by hypothesis, every $R_\ell(G_0)$ have no off-diagonal positives. Inductively this shows that every column of $h_\ell(G_0)$ has support in exactly one row. We define the label of this column to be the row index $\in \{1, \ldots, n\}$ where the unique support is. With labels defined, the remaining proof is based on establishing the following locality claim.

CLAIM. Fix graph $G$, layer $\ell$, and $i, j \in \{1, \ldots, n\}$. If column $c$ of $h_\ell(G)$ has label $j$ and if $h_\ell(G)_{i,c} > 0$, then $\operatorname{dist}_G(i, j) \leq 3^\ell$. In other words, *every column spreads at most $3^\ell$ hops away from its label by depth $\ell$.*

STEP 2. ESTABLISHING THE CLAIM. We prove this claim via induction. The base case $\ell = 0$ directly follows from the fact that $h_0(G) = [I_n \mid A(G)]$. For the inductive step, we assume that the claim holds at depth $\ell - 1$ with radius $3^{\ell-1}$. As in Lemma B.1, there are two column types in $h_\ell$: inherited or newly appended columns. The former case is easy; if $c$ is inherited from $h_{\ell-1}$, then

$h_\ell(G)_{i,c} = h_{\ell-1}(G)_{i,c}$, so the bound follows from the inductive hypothesis. We now assume $c$ is newly appended.

Suppose $(R_\ell(G)h_{\ell-1}(G))_{i,c} > 0$ for a column $c$ with label $j$. Then there exists a row $k$ with $R_\ell(G)_{i,k} > 0$ and $h_{\ell-1}(G)_{k,c} > 0$. By the IH, $\mathrm{dist}_G(k,j) \leq 3^{\ell-1}$. Then we expand $R_\ell(G)_{i,k} > 0$ to obtain column witnesses $p, q$, with $h_{\ell-1}(G)_{i,p} > 0$, $h_{\ell-1}(G)_{k,q} > 0$, and $(W_\ell)_{p,q} > 0$, as in Lemma B.1. Let $a, b$ be the labels of $p, q$, respectively. By IH again, $\mathrm{dist}_G(i,a) \leq 3^{\ell-1}$ and $\mathrm{dist}_G(k,b) \leq 3^{\ell-1}$. We now split the analysis into two cases.

- If $a \neq b$, we derive a contradiction to the no-false-positives assumption by reusing the certificate procedure from Lemma B.1. Because $W_\ell \geq 0$ entrywise, every positive entry in $h_t(\cdot)$ admits a certificate supported on at most $s(t) \leq 2 \cdot 3^t$ vertices. In particular, there exist certificates witnessing $h_{\ell-1}(G)_{i,p} > 0$ (labeled $a$) and $h_{\ell-1}(G)_{k,q} > 0$ (labeled $b$). Let $S_a, S_b$ be the corresponding certificate vertex sets. Form a new graph $G'$ on the same $n$ vertices whose connected components are two disjoint induced copies $S'_a, S'_b$ of the subgraphs on $S_a, S_b$ (leaving all other vertices outside $S_a \cup S_b$ isolated). Note this is feasible because $|S_a| + |S_b| \leq 4 \cdot 3^{L-1} \leq n$ assumed by Theorem 4.4. By construction, there exist $i' \in S'_a$ and $k' \in S'_b$ with $h_{\ell-1}(G')_{i',p} > 0$ and $h_{\ell-1}(G')_{k',q} > 0$. Thus,

$$(h_{\ell-1}(G')W_\ell h_{\ell-1}(G')^\top)_{i',k'} \geq h_{\ell-1}(G')_{i',p}(W_\ell)_{p,q}h_{\ell-1}(G')_{k',q} > 0,$$

  meaning $R_\ell(G')_{i',k'} > 0$. But $i', k'$ belong to different connected components in $G'$, contradiction! Therefore,

- $a = b$. Triangle inequality gives $\mathrm{dist}_G(i,j) \leq \mathrm{dist}_G(i,a) + \mathrm{dist}_G(a,k) + \mathrm{dist}_G(k,j) \leq 3 \cdot 3^{\ell-1} = 3^\ell$, completing the induction.      END PROOF OF CLAIM / STEP 2.

The model's output $\mathsf{TF}_\Theta^L(A(G)) = h_L(G)W_O^\top$ is an entrywise nonnegative sum over the $n \times n$ blocks of $h_L(G)$. Since each block respects the $3^L$ locality bound, we have $\mathsf{TF}_\Theta^L(A(G))_{u,v} = 0$ whenever $\mathrm{dist}_G(u,v) > 3^L$. Hence, on any graph whose largest component has diameter $> 3^L$, the model will inevitably miss a pair $(u,v)$ of nodes realizing this diameter.     □

*Proof of Theorem 4.4.* Combine Lemmas B.1 and B.2.     □

## C    DETAILS FOR TRAINING DYNAMICS

### C.1    CHARACTERIZING BLOCK WEIGHTS $W_\ell$

As discussed in Section 4.3, due to the symmetric nature of the graph connectivity problem, it is natural to demand that a "good" model should map not only adjacency matrices $A$ to connectivity matrices $R$, but also $PAP^\top$ to $PRP^\top$ for any permutation $P$. We further generalize equivariance. Observe that given a permutation matrix $P$ and any hidden states $h \in \mathbb{R}^{n \times (kn)}$ consisting of $k$ consecutive $n \times n$, the mapping $h \mapsto Ph(I_K \otimes P^\top)$ relabels both rows and columns within each $n \times n$ block in a way that is consistent with the effects of $P$. Hence, the notion of equivariance can be generalized to any (nonnegative) hidden states, beyond just the ones induced by adjacency matrices.

Similarly, we are now also able to define an $L$-layer disentangled transformer on arbitrary inputs of appropriate dimensions. For any nonnegative initial state $h_0 \in \mathbb{R}^{n \times 2n}$, recursively define $h_\ell = [h_{\ell-1} \mid \mathrm{Attn}(h_{\ell-1}; W_\ell)]$ for $\ell = 1, \dots, L$. Let $\mathrm{Sum}(h)$ denote the sum of the consecutive left-aligned $n \times n$ blocks of $h$. Then the generalized output is $\mathsf{TF}_\Theta^L(h_0) = \mathrm{Sum}(h_L)$. We define two equivariance-related conditions. The first one is a direct generalization of $P\mathsf{TF}_\Theta^L(A)P^\top = \mathsf{TF}_\Theta^L(PAP^\top)$; the second one, as discussed in Section 4.3, makes theoretical analysis significantly more tractable while also being supported by empirical evidence.

**Definition C.1** (Output Equivariance and Layerwise Attention Equivariance)**.** *Let $\mathsf{TF}_\Theta^L$ be an $L$-layer disentangled transformer with nonnegative weights. Let $K_\ell = 2^{\ell+1}$.*

*(i)* *For $h_0 \in \mathbb{R}_{\geq 0}^{n \times 2n}$ and for any $P$, define $h_0^P = Ph_0(I_{K_0} \otimes P^\top)$. We say $\mathsf{TF}_\Theta^L$ is **output-level value equivariant** iff $P\mathsf{TF}_\Theta^L(h_0)P^\top = \mathsf{TF}_\Theta^L(h_0^P)$ holds for all $P$ and all $h_0 \in \mathbb{R}_{\geq 0}^{n \times 2n}$.*

*(ii)* *We say $\mathsf{TF}_\Theta^L$ is **layer-wise attention equivariant** iff for each $\ell$ and any hidden states $h \in \mathbb{R}^{n \times d_{\ell-1}}$ (i.e., any hidden states of dimension feasible for layer $\ell$),*

$$\mathrm{Attn}(Ph(I_{K_{\ell-1}} \otimes P^\top); W_\ell) = P\,\mathrm{Attn}(h; W_\ell)\,(I_{K_{\ell-1}} \otimes P^\top),$$

**Theorem C.2** (Parameterization of "Good" Models)**.** *Let $n = \Omega(3^L)$ as in Theorem 4.4. Fix an $L$-layer Disentangled Transformer $\mathsf{TF}_\Theta^L$ with nonnegative weights. Suppose that*

*(i)* $\mathsf{TF}_\Theta^L$ *is output-level value-equivariant, and*

*(ii)* $\mathsf{TF}_\Theta^L$ *reaches its capacity bound of $3^L$, i.e., for every graph, we have $\mathrm{supp}(\mathsf{TF}_\Theta^L(A)) = \mathrm{supp}(A^{3^L})$.*

*Then, either $\mathsf{TF}_\Theta^L$ or a functionally equivalent version of it satisfies the following: for each layer $\ell$, there exists a nonnegative matrix $\Lambda_\ell \in \mathbb{R}^{K_{\ell-1} \times K_{\ell-1}}$ such that $W_\ell + W_\ell^\top = \Lambda_\ell \otimes I_n$. In other words, $W_\ell$ can be decomposed into this form up to an antisymmetric part.*

*(Note that the theorem is a direct generalization of equivariance under all graph permutations; replacing $h_0$ by $[I_n \mid A]$ gives the desired result for a fixed graph with adjacency matrix $A$.)*

*Proof.* To prove the claim, it suffices to show that if we partition $W_\ell$ into $K_{\ell-1} \times K_{\ell-1}$ contiguous sub-blocks of size $n \times n$, then each block must be diagonal, with symmetry conditions meeting $W_\ell + W_\ell^\top = \Lambda_\ell \otimes I_n$.

To do so, the proof is split into two parts: we prove that each $n \times n$ block must be diagonal using (ii) and Lemma B.2, and that the diagonal entries must realize the said forms by examining the forward maps under a curated, parameterized class of initial hidden states.

STEP 1: EACH BLOCK MUST BE DIAGONAL. In this step, we argue that if a block admits a positive off-diagonal entry, then the certificate trick from Lemma B.1 will create a false positive entry on some output, contradicting (ii).

Formally, let $R_\ell = \mathsf{ReLU}(h_{\ell-1}W_\ell h_{\ell-1}^\top)$. If for some graph and some $\ell$, there exists a false positive entry $(R_\ell)_{i,k} > 0$ for some $i, k$ across different connected components, then the false positiveness would persist to the output, contradicting (ii). Hence $\mathsf{TF}_\Theta^L$ must have no false positives.

Consider feeding the graph $G_0$ of $n$ isolated vertices into $\mathsf{TF}_\Theta^L$, so that $h_0(G_0) = [I_n \mid I_n]$. The premises of Lemma B.2 hold, so every column of every $h_\ell(G_0)$ is supported in exactly one row, which we called its label in $\{1, \ldots, n\}$. Hence, if we write $h_\ell(G_0) = [X_1^{(\ell)} \mid \ldots \mid X_{K_\ell}^{(\ell)}]$ of contiguous $n \times n$ blocks, then each such block $X_r^{(\ell)}$ must be nonnegative and diagonal. Now expand

$$R_\ell = \mathsf{ReLU}(h_{\ell-1} W_\ell h_{\ell-1}^\top) = h_{\ell-1} W_\ell h_{\ell-1}^\top = \sum_{r,s} X_r^{(\ell-1)} W_\ell[r,s](X_s^{(\ell-1)})^\top.$$

We first claim that every $n \times n$ sub-block $W_\ell[r,s]$ is diagonal. Suppose not, that there exist indices $r, s$ and distinct nodes $i \neq k$ such that $(W_\ell[r,s])_{i,k} > 0$. For a node $i$ and a block $r$, we say $(i,r)$ is *activatable* at depth $\ell - 1$ if there exists *some* graph $G$ such that $X_r^{(\ell-1)}(G)[i,i] = h_{\ell-1}[i, (r-1)n+i] > 0$. Two cases:

- If at least one of $(i,r)$ or $(k,s)$ is not activatable, then for every graph $G$, at least one factor $X_r^{(\ell-1)}(G)[i,i]$ or $X_s^{(\ell-1)}(G)[k,k]$ is zero, and thus $(W_\ell[r,s])_{i,k}$ is functionally inert and never contributes to any $R_\ell$ entry. Hence we may simply set it to 0 without altering the model's output on any graph.

- If both $(i,r)$ and $(k,s)$ are activatable, take graphs $G_i, G_k$ that make $X_r^{(\ell-1)}(G_i)[i,i] > 0$ and $X_x^{(\ell-1)}(G_k)[k,k] > 0$. Using the certificate mechanism in Lemma B.1, each positiveness admits a finite certificate subgraph with at most $2 \cdot 3^{\ell-1}$ vertices. We then create a new graph $G'$ and disjointly embed both certificates into it, leaving all other vertex isolated. The two labels $i, k$, viewed as nodes, now lie in different components. But then the product

$$X_r(G')[i,i] \cdot (W_\ell[r,s])_{i,k} \cdot X_k(G')[k,k] > 0,$$

making $(R_\ell(G'))_{i,k} > 0$, contradiction.

Therefore $W_\ell[r,s]$ is diagonal for all block indices $(r,s)$. This concludes STEP 1.

STEP 2. $W_\ell$ IS NODE-SYMMETRIC. Given a triplet $(\ell, r, s)$, we can now write $W_\ell[r,s]$ as $\mathrm{diag}(w_{\ell,r,s}(1), \ldots, w_{\ell,r,s}(n))$. Our goal is to show that for each $(\ell, r, s)$, $w_{\ell,r,s}(j) + w_{\ell,s,r}(j) = w_{\ell,r,s}(k) + w_{\ell,s,r}(k)$ for all $j, k \in [n]$. We formalize this in matrix form: For each node $i \in [n]$ and each layer $\ell$, let $\Lambda_\ell^{(i)} = [w_{\ell,r,s}(i)]_{r,s} \in \mathbb{R}^{K_{\ell-1} \times K_{\ell-1}}$ and define the symmetric part $\mathrm{Sym}(\Lambda_\ell^{(i)}) = (\Lambda_\ell^{(i)} + \Lambda_\ell^{(i)T})/2$; the goal is to show that given $\ell$, all $\Lambda_\ell^{(i)}$ are the same, so that $\mathrm{Sym}(W_\ell) = \Lambda_\ell \otimes I_n$ or equivalently, $W_\ell + W_\ell^\top = \Lambda_\ell \otimes I_n$, as claimed.

Throughout out this step, we will use a family of special hidden states parameterized by a scalar $\lambda > 0$ and a vector $u = (u_1, u_2) \in \mathbb{R}_{\geq 0}^2$. Fix distinct nodes $j \neq k$. For $\lambda, u$, define the initial state $h_0(\lambda, u) \in \mathbb{R}^{n \times 2n}$ by setting exactly four entries nonzero:

$$\begin{cases} h_0(\lambda, u)[j,j] = \lambda u_1 & h_0(\lambda, u)[j, n+j] = \lambda u_2 \\ h_0(\lambda, u)[k,k] = \lambda u_1 & h_0(\lambda, u)[k, n+k] = \lambda u_2. \end{cases}$$

Note that $h_0(\lambda, u)$ is invariant under the transposition $P = (j,k)$, i.e., $P h_0(\lambda, u)(I_{K_0} \otimes P^\top) = h_0(\lambda, u)$. Therefore, by assumption (i), we must have $\mathsf{TF}_\Theta^L(h_0)_{j,j} = \mathsf{TF}_\Theta^L(h_0)_{k,k}$. Let $h_\ell(\lambda, u)$ be the network state at depth $\ell$. Because of STEP 1, there is no cross-row interaction for this input at any depth. Writing the row-$i$ vector as $v_\ell^{(i)}(\lambda, u) \in \mathbb{R}^{K_\ell}$, recursion gives, for $i \in \{j, k\}$,

$$v_0^{(i)}(\lambda, u) = \lambda u, \qquad v_\ell^{(i)}(\lambda, u) = [v_{\ell-1}^{(i)}(\lambda, u) \mid q_\ell^{(i)}(\lambda, u) v_{\ell-1}^{(i)}(\lambda, u)]$$

where

$$q_\ell^{(i)}(\lambda, u) = \frac{1}{n} \cdot v_{\ell-1}^{(i)}(\lambda, u)^\top \, \mathrm{Sym}(\Lambda_\ell^{(i)}) \, v_{\ell-1}^{(i)}(\lambda, u).$$

Taking $\ell_1$-norms gives

$$\|v_\ell^{(i)}(\lambda, u)\| = (1 + q_\ell^{(i)}(\lambda, u))\|v_{\ell-1}^{(i)}(\lambda, u)\| \quad \text{and} \quad \|v_L^{(i)}(\lambda, u)\| = \|v_0^{(i)}(\lambda, u)\| \prod_{\ell=1}^L (1 + q_\ell^{(i)}(\lambda, u)).$$

$$(7)$$

Because the readout weight $W_O$ is a concatenation of $I_n$'s, and under our specific input $h_0(\lambda, u)$, every nonzero row $i$ lies in columns with indices $i$ modulo $n$, the $(i, i)$ output numerically equals $\|v_L^{(i)}(\lambda, u)\|$. Hence, assumption (i) requires $\|v_L^{(j)}(\lambda, u)\| = \|v_L^{(k)}(\lambda, u)\|$.

Let $\ell^*$ be the minimal layer such that $\text{Sym}(\Lambda_{\ell^*}^{(j)}) \neq \text{Sym}(\Lambda_{\ell^*}^{(k)})$. If no such $\ell^*$ exists for all $j \neq k$, then all $\Lambda_\ell^{(i)}$'s are the same given any fixed $\ell$, and STEP 2 holds. Otherwise, for every $\ell < \ell^*$, the symmetric parts coincide, and $v_{\ell-1}^{(j)}(\lambda, u) = v_{\ell-1}^{(k)}(\lambda, u)$ and $q_\ell^{(j)}(\lambda, u) = q_\ell^{(k)}(\lambda, u)$ for all $\lambda, u$. We may use $v_{\ell^*-1}(\lambda, u)$ to denote both $v_{\ell^*-1}^{(j)}(\lambda, u)$ and $v_{\ell^*-1}^{(k)}(\lambda, u)$ for they are now equal.

Because of the structure of $h_0(\lambda, u)$, by induction, the row vectors of each hidden state admits an odd power expansion

$$v_{\ell-1}^{(i)}(\lambda, u) = \lambda u + \lambda^3 \xi_{1,\ell-1}(u) + \lambda^5 \xi_{2,\ell-1}(u) + \dots$$

from which we conclude $q_\ell^{(i)}(\lambda, u) = O(\lambda^2)$ for every $\ell$. In particular, at $\ell = \ell^*$,

$$q_{\ell^*}^{(j)}(\lambda, u) - q_{\ell^*}^{(k)}(\lambda, u) = \frac{1}{n} \cdot v_{\ell^*-1}(\lambda, u)^\top (\text{Sym}(\Lambda_{\ell^*}^{(j)}) - \text{Sym}(\Lambda_{\ell^*}^{(k)})) v_{\ell^*-1}(\lambda, u) = \lambda^{2m} c(u) + o(\lambda^{2m})$$

for some $m \geq 1$ and some nondegenerate polynomial $c(u)$ as $\lambda \searrow 0$. In particular,

$$q_{\ell^*}^{(j)}(\lambda, u) - q_{\ell^*}^{(k)}(\lambda, u) = \Theta(\lambda^{2m}).$$

We now put this back into the comparison between the output's $(j, j)$ and $(k, k)$ entry. Recall that $v_0^{(j)}(\lambda, u) = v_0^{(k)}(\lambda, u) = \lambda u$. Further, since $v_{\ell-1} = \lambda u + O(\lambda^3)$, we know $q_\ell(\lambda, u) = O(\lambda^2)$ for every $\ell$ and every $i$. We drop $\lambda, u$ for notational simplicity. It follows from equation 7 that

$$\|v_L^{(j)}\| - \|v_L^{(k)}\| = \lambda \|u\| \cdot \left[ \prod_{l < \ell^*} (1 + q_\ell) \right] \cdot \left[ \left[ 1 + q_{\ell^*}^{(j)} \right] \prod_{\ell > \ell^*} \left[ 1 + q_\ell^{(j)} \right] - \left[ 1 + q_{\ell^*}^{(k)} \right] \prod_{\ell > \ell^*} \left[ 1 + q_\ell^{(k)} \right] \right]$$

$$= \lambda \|u\| \cdot \left[ \prod_{\ell < \ell^*} (1 + O(\lambda^2)) \right] \cdot \left[ q_{\ell^*}^{(j)} - q_{\ell^*}^{(k)} \right] \cdot \left[ \prod_{\ell > \ell^*} (1 + O(\lambda^2)) \right]$$

$$= \lambda \|u\| \Theta(\lambda^{2m})(1 + o(1)) = \Theta(\lambda^{2m+1})$$

which is nonzero for small $\lambda$. Hence the $(j, j)$ and $(k, k)$ entries can be made different, contradicting assumption (i), and the proof is complete!  $\square$

**Theorem 4.6.** Suppose an $L$-layer Disentangled Transformer $\text{TF}_\Theta^L$ has nonnegative parameters. Suppose $\text{TF}_\Theta^L$ is layerwise permutation equivariant, i.e., for each $\ell$ and any hidden states $h \in \mathbb{R}^{n \times d_{\ell-1}}$,

$$\text{Attn}(Ph(I_{K_{\ell-1}} \otimes P^\top); W_\ell) = P \, \text{Attn}(h; W_\ell) \, (I_{K_{\ell-1}} \otimes P^\top),$$

then each block $W_\ell = A_\ell \otimes I_n + B_\ell \otimes J_n$ for some $A_\ell, B_\ell \in \mathbb{R}^{K_{\ell-1}, K_{\ell-1}}$. In other words, each block-aligned $n \times n$ submatrix of $W_\ell$ necessarily lies in $\text{span}\{I_n, J_n\}$.

**Remark C.3.** The equivariance condition presented in the theorem is strictly harder than what we need for graph-level, layerwise equivariance:

$$\text{Attn}(h_{\ell-1}(PAP^\top); W_\ell) = P \, \text{Attn}(h_{\ell-1}(A); W_\ell) \, (I_{K_\ell-1} \otimes P^\top).$$

For graphs, it suffices to assume that the hidden states are induced by some $n$-node graph.

*Proof.* STEP 1. RELATING TO WEIGHT CONJUGATION. Fix a layer $\ell$. Write $K = K_{\ell-1}$, $h = h_\ell$, $W = W_\ell$, and let $\sigma(P) = I_K \otimes P$. The first step is to relate the conjugation of hidden states, $T_P(h) : h \mapsto Ph(I_K \otimes P^\top)$, to a conjugation of layer weights, $W_\ell \mapsto \sigma(P)W_\ell\sigma(P)^\top$.

Concretely, since $W_\ell \geq 0$, ReLU. Hence

$$\text{Attn}(T_P(h); W) = \frac{1}{n} \text{ReLU}[(Ph\sigma(P)) \, W \, (\sigma(P)^\top h^\top P^\top)](Ph\sigma(P))$$

$$= \frac{1}{n} P[h\sigma(P) \, W \, (\sigma(P)^\top h^\top)](h\sigma(P))$$

and

$$T_P(\text{Attn}(h; W)) = \frac{1}{n} P(hWh^\top) h \sigma(P).$$

Layer-wise attention equivariance requires the two quantities above to equal for all $h$, and left multiplication by $P^{-1}$ gives

$$h\Delta h^\top h\, \sigma(P) = 0 \qquad \text{for all } h \geq 0 \qquad \text{where} \qquad \Delta := \sigma(P)W\sigma(P)^\top - W. \qquad (*)$$

STEP 2. PROVING $\Delta = 0$. To do so, we consider special hidden states, with only two nonzero entries $h_{i,p} = 1$ and $h_{j,q} = t$. Equivalently, pick columns $p \neq q$ and rows/nodes $i \neq k$ and set $h_{i,.} = e_p^\top$, $h_{j,.} = t e_q^\top$, and $h = 0$ everywhere else, where $e_p$ is standard basis vector pivoted at $p$.

Because $h$ only uses columns $p$ and $q$, the matrix $h\Delta h^\top$ can be embedded on rows/columns $\{i, j\}$ with values

$$h\Delta h^\top = \begin{pmatrix} \Delta_{p,p} & t\Delta_{p,q} \\ t\Delta_{q,p} & t^2\Delta_{q,q} \end{pmatrix}.$$

Recall $\sigma(P)$ is a permutation on columns; let $\pi$ be the permutation induced by it. Since $h\sigma(P)$ has the same two nonzero rows with $(h\sigma(P))_{i,.} = e_{\pi(p)}^\top$ and $(h\sigma(P))_{j,.} = t e_{\pi(q)}^\top$, we get that $(h\Delta h^\top)(h\sigma(P))$ only has rows $i$ and $j$ potentially nonzero:

$$\begin{cases} \text{row } i : \Delta_{p,p} e_{\pi(p)}^\top + t^2 \Delta_{p,q} e_{\pi(q)}^\top \\ \text{row } j : t\Delta_{q,p} e_{\pi(p)}^\top + t^2 \Delta_{q,q} e_{\pi(q)}^\top. \end{cases}$$

But recall (*): $(h\Delta h^\top)(h\sigma(P)) = 0$ for all $t > 0$. The two standard basis vectors $e_{\pi(p)}, e_{\pi(q)}$ are linearly independent, so the coefficients must be uniformly zero! Hence $\Delta_{p,p} = \Delta_{p,q} = \Delta_{q,p} = \Delta_{q,q} = 0$. Finally, because $p \neq q$ were arbitrary, this forces $\Delta = 0$ entrywise. and that $\sigma(P)W_\ell\sigma(P)^\top = W_\ell$ for this $P$. And because $P$ is arbitrary, we conclude that $\sigma(P)W\sigma(P)^\top = W$ for every permutation $P$.

STEP 3. RELATING TO $n \times n$ BLOCKS. Consider any $n \times n$ block $W[u,v]$ of $W$ where $1 \leq u, v \leq K_\ell$. Using $\sigma(P) = I_{K_\ell} \otimes P^\top$ and taking the $(u,v)$ block on both sides,

$$(\sigma(P)W\sigma(P)^\top)[u,v] = \sum_{a,b} (I_{K_\ell})_{u,a} P^\top W[a,b] P(I_{K_\ell})_{b,v} = P^\top W[u,v]P.$$

The LHS equals $W[u,v]$, so we conclude that

$$P^\top W[u,v]P = W[u,v] \qquad \text{for all } P \in S_n.$$

In other words, layerwise equivariance implies each block must be invariant under $P^\top(\cdot)P$. Taking any transposition forces all diagonal entries of a block to equal, while for any $i \neq j, k \neq \ell$, any arbitrary permutation mapping $\pi(i) = k, \pi(j) = \ell$ forces entries $(i,j)$ and $(k,\ell)$ to be equal. This implies that each block lies in $\text{span}\{I_n, J_n\}$ as claimed. $\qquad\square$

## C.2  POPULATION GRADIENT LIVES IN THE EQUIVARIANT ALGEBRA

**Theorem C.4** (Population gradient lives in the equivariant algebra). *Under Assumption 4.7, in particular using layerwise parameterization $W_\ell = A_\ell \otimes I_n + B_\ell \otimes J_n$, fix a layer $\ell$ and let $K = K_{\ell-1}$. Then the population gradient with respect to $W_\ell$ lies in $M_K(\mathbb{R}) \otimes \text{span}\{I_n, J_n\}$: there exist matrices $G_\ell^{(I)}, G_\ell^{(J)} \in \mathbb{R}^{K \times K}$ such that*

$$\mathbb{E}\Big[\frac{\partial \mathcal{L}}{\partial W_\ell}\Big] = G_\ell^{(I)} \otimes I_n + G_\ell^{(J)} \otimes J_n. \tag{8}$$

*Proof.* We let $S_n$ act on node indices. Since $W_\ell$ can be parametrized as $W_\ell = A_\ell \otimes I_n + B_\ell \otimes J_n$, the attention map is equivariant under left-right action:

$$\text{Attn}(Ph(I_K \otimes P^\top); W_\ell) = P\,\text{Attn}(h; W_\ell)(I_K \otimes P^\top),$$

and so is the full map $A \mapsto Z$. For any fixed permutation $P$, the data $\mathsf{ER}(n, p)$ is permutation-invariant, i.e., $A$ and $PAP^\top$ are identically distributed. Because the model map and the loss are equivariant under $A \mapsto PAP^\top$ with $R \mapsto PRP^\top$, the sample gradient covaries as

$$\nabla_{W_\ell}\mathcal{L}(PAP^\top) = (I_k \otimes P)\nabla_{W_\ell}\mathcal{L}(A)(I_K \otimes P^\top).$$

Taking expectation over $A$ gives

$$\mathbb{E}_A[\nabla_{W_\ell}\mathcal{L}(PAP^\top)] = (I_k \otimes P)\mathbb{E}_A[\nabla_{W_\ell}\mathcal{L}(A)](I_K \otimes P^\top)$$

for every $P$. Hence the population gradient lies in the commutant of $\{I_K \otimes P : P \in S_n\}$. It remains to identify this commutant. View $G_\ell$ as a $K \times K$ block matrix with $n \times n$ sub-blocks. The relation $(I_K \otimes P)^\top G_\ell(I_K \otimes P) = G_\ell$ says each $n \times n$ block $B$ satisfies $P^\top B P$ for all permutations $P$, so the block must have one value on the diagonal and one on the off-diagonals. It is well known that the fixed-point algebra of conjugation on $n \times n$ matrices is $\mathrm{span}(I_n, J_n)$. Hence every block lies in this span, i.e., $G_\ell \in M_K(\mathbb{R}) \otimes \mathrm{span}\{I_n, J_n\}$. $\qquad\square$

### C.3 WHICH CONDITIONS ENCOURAGE $W_\ell \approx A_\ell \otimes I_n$?

To facilitate the following analyses, it will be beneficial to first (re)introduce some notations.

Throughout the analysis of training dynamics, we inherit the notations used in Assumption 4.7: we use $Z$ to denote the model output, $R$ the reachability matrix, $A$ the adjacency matrix, $\mathcal{L} = \mathcal{L}(Z; R)$ the loss, and $\mathcal{R}(\Theta)$ the population risk $\mathcal{R}(\Theta) := \mathbb{E}_{G \sim \mathsf{ER}(n,p)}[\mathcal{L}(\mathsf{TF}_\Theta^L(A_G); R_G)]$.

Fix a layer $\ell$ and a nonnegative direction $\Delta \geq 0$ in the $J$-channel. Write $D = \frac{\partial Z}{\partial B_\ell}[\Delta]$ (more details in Theorem C.5). We say a node pair $(i, j)$ is **active** for $\Delta$ if $D_{i,j} > 0$. In particular, we say $\Delta$ is active on cross-component pairs if $D_{i,j} > 0$ for some $(i, j)$ belonging to different connected components (note $\Delta$ could also be active on within-component pairs).

Because we constrain $W_\ell \geq 0$, under the parameterization $W_\ell = A_\ell \otimes I_n + B_\ell \otimes J_n$, we must also have $B_\ell \geq 0$. Then, the appropriate notion of stationarity is KKT: in our setting, this reduces to

$$\nabla_{B_\ell}\mathcal{R}(\Theta) \geq 0, \qquad B_\ell \geq 0, \qquad \text{and} \qquad \nabla_{B_\ell}\mathcal{R}(\Theta) \odot B_\ell = 0$$

which we use in the Theorem below.

**Theorem C.5** (Population Training Conditionally Suppresses the $J$-Channel). *Assume Assumption 4.7. Fix any layer $\ell$ and decompose $W_\ell = A_\ell \otimes I_n + B_\ell \otimes J_n$. Let $Z$ be the output, $R$ the reachability matrix (ground truth), $\mathcal{L} = \mathcal{L}(Z; R)$ the loss, and $\mathcal{R}(\Theta)$ the population risk.*

*1. (**Directional derivative on nonnegative $J$-channel directions**.) Let $\Delta \in \mathbb{R}^{K_{\ell-1} \times K_{\ell-1}}$ be entrywise nonnegative and define the one-sided Fréchet derivative $D := \frac{\partial Z}{\partial B_\ell}[\Delta] \in \mathbb{R}^{n \times n}$. Then $D \geq 0$ entrywise, and the population directional derivative satisfies*

$$D_{B_\ell}\mathcal{R}(\Theta)[\Delta] = \mathbb{E}\left\langle \left[\frac{\partial \mathcal{L}}{\partial Z}, D\right]\right\rangle_F = \alpha \cdot \mathbb{E}\left[\underbrace{\sum_{R_{i,j}=0} D_{i,j}}_{\substack{\text{penalty from} \\ \text{cross component}}} - \underbrace{\sum_{R_{i,j}=1} \frac{1 - \phi_\epsilon(Z_{i,j})}{\phi_\epsilon(Z_{i,j})}D_{i,j}}_{\substack{\text{weighted reward on} \\ \text{under-predicted positives}}}\right]. \quad (9)$$

*In particular, $D_{B_\ell}\mathcal{R}(\Theta)[\Delta] \geq 0$ iff **cross-component penalty** $\geq$ **within-component reward**. Throughout this section, we will be using these names to denote the two competing sums whenever an expression like equation 9 appears.*

*2. (**Consequences for KKT stationary points**.) Assume $\Theta$ is KKT-stationary for $B_\ell \geq 0$:*

$$\nabla_{B_\ell}\mathcal{R}(\Theta) \geq 0, \qquad B_\ell \geq 0, \qquad \text{and} \qquad \nabla_{B_\ell}\mathcal{R}(\Theta) \odot B_\ell = 0 \quad (10)$$

*Let $\Delta = |B_\ell|$ (entrywise absolute value) and let $D = \frac{\partial Z}{\partial B_\ell}[|B_\ell|]$. If, with positive probability under $\mathsf{ER}(n, p)$, $\Delta$ activates at least one cross-component pair, and if the cross component penalty term strictly dominates the within-component reward, then $B_\ell = 0$. Equivalently, under activation at $\Delta = |B_\ell|$ and strict dominance by cross-component penalty, the only KKT stationary point in the $J_n$-channel is $B_\ell = 0$.*

**Lemma C.6** (Monotonicity in the $J$-channel). *Fix $\ell$ and hold all parameters except $B_\ell$. Write $h_{\ell-1} = [X_1 \mid \ldots \mid X_{K_{\ell-1}}]$ and $u_p = X_p \boldsymbol{1} \in \mathbb{R}^n_{\geq 0}$. Then*

$$h_{\ell-1} W_\ell h_{\ell-1}^\top = \sum_{p,q} (A_\ell)_{p,q} X_p X_q^\top + \sum_{p,q} (B_\ell)_{p,q} u_p u_q^\top. \tag{11}$$

*Consequently, for every nonnegative direction $\Delta \geq 0$ in the $J$-channel, the one-sided Fréchet derivative at $0^+$ exists and is entrywise nonnegative. Hence, along the ray $\{B_\ell + \delta\Delta \mid \delta \geq 0\}$, the output is entrywise nondecreasing:*

$$\frac{\partial Z}{\partial B_\ell}[\Delta] \in \mathbb{R}^{n \times n}_{\geq 0}, \qquad Z(B_\ell + \delta\Delta) - Z(B_\ell) \geq 0 \text{ for all } \delta \geq 0.$$

*Moreover, if $G$ is disconnected, and either (i) $\Delta_{p,p} > 0$ for a block $p$ such that $u_p$ has support in at least two components, or (ii) there exist blocks $p, q$ with $\Delta_{p,q} > 0$ and $u_p, u_q$ supported in different components, then there exist cross component pairs $(i, j)$ with $(\frac{\partial Z}{\partial B_\ell}[\Delta])_{i,j} > 0$.*

*Proof.* Since $J_n x = (\boldsymbol{1}^\top x)\boldsymbol{1}$ for $x \in \mathbb{R}^n$, we have $X_p J_n X_q^\top = (X_p \boldsymbol{1})(X_q \boldsymbol{1})^\top = u_p u_q^\top$, yielding the displayed decomposition. For $B_\ell \mapsto B_\ell + \delta\Delta$ with $\Delta \geq 0$, the layer scores

$$R_\ell(B_\ell + \delta\Delta) - R_\ell(B_\ell) = \delta \sum_{p,q} \Delta_{p,q} u_p u_q^\top \geq 0,$$

so the one-sided derivative exists and is entrywise nonnegative. Because all subsequent maps are entrywise monotone, this implies $Z(B_\ell + \delta\Delta) - Z(B_\ell) \geq 0$ as stated.

For the "moreover" part, in casse (i), $u_p u_p^\top$ places positive mass on index pairs spnaning the componentns where $u_p > 0$, and in case (ii), $u_p u_q^\top$ (or its transpose) places positive mass across two components supporting $u_p$ and $u_q$. Monotonicity propagates these positives to $D = \frac{\partial Z}{\partial B_\ell}[\Delta]$. $\qquad\square$

*Proof of Theorem C.5.* For the population risk $\mathcal{R}(\Theta) = \mathbb{E}[\mathcal{L}(Z; R)]$, applying definitions gives the directional derivative along $\Delta$ gives

$$D_{B_\ell} \mathcal{R}(\Theta)[\Delta] = \left\langle \mathbb{E}\Big[\frac{\partial \mathcal{L}}{\partial Z}\Big], D \right\rangle_F = \alpha \cdot \mathbb{E}\Big[ \sum_{i,j} \Big(1 - \frac{R_{i,j}}{\phi_\epsilon(Z_{i,j})}\Big) D_{i,j}\Big].$$

Separating indices by $R_{i,j} \in \{0, 1\}$ proves equation 9.

For the second claim, evaluate equation 9 at $\Delta = |B_\ell|$. Under the activation premise (Lemma C.6) and strict dominance by cross-component penalty, we obtain $D_{B_\ell} \mathcal{R}(\Theta)[|B_\ell|] = \langle \nabla_{B_\ell} \mathcal{R}(\Theta), \Delta \rangle_F > 0$. Since $\nabla_{B_\ell} \mathcal{R}(\Theta) \geq 0$ and $|B_\ell| \geq 0$, a strictly positive inner product violates the KKT complementary condition $\nabla_{B_\ell} \mathcal{R}(\Theta) \odot B_\ell = 0$ unless $B_\ell = 0$. $\qquad\square$

**Remark C.7.** While Theorem C.5 mostly discusses the suppression of $B_\ell$, its (i) in fact reveals a quite interesting, opposite phenomenon: **early training promotes** $B_\ell$. Before the model learns to pick up easy connected pairs, the corresponding values $\phi_\epsilon(Z_{i,j}) \ll 1$. Consequently, the fractions $(1 - \phi_\epsilon(Z_{i,j}))/\phi_\epsilon(Z_{i,j})$ are large, making equation 9 negative. Gradient descent then pushes $B_\ell$ up "without feeling pressure." As training proceeds, these easy connected pairs saturate ($\phi_\epsilon(Z_{i,j}) \to 1$), while simultaneously $\Delta$ begins to active cross pairs (the "moreover" part of Lemma C.6), increasing the $R = 0$ term in equation 9 and potentially flipping the sign. This is when the $J$-channel starts to incur penalty. This explains the transient "Phase 1" in §4.3.

**Remark C.8.** The $B_\ell \otimes J_n$-channel injects rank-one dense terms $u_p u_q^\top$ into the attention core. On disconnected graphs, these terms produce cross-component positives, which the reachability target $R$ labels as negatives. Because disconnected graphs appear with positive probability in the data, the population gradient penalizes every nonnegative direction in the $J$-channel active on cross-component pairs whenever the cross-component penalty dominates within-component reward. Under the same activation and cross-component penalty dominance assumptions, any KKT stationary point must have $B_\ell = 0$. In short: under these conditions, population drives the node-side factor towards locality, i.e., $W_\ell \approx A_\ell \otimes I_n$.

## C.4 Which Samples Push Which Channel? (Local $I_n$ vs. Global $J_n$)

Recall $W_\ell = A_\ell \otimes I_n + B_\ell \otimes J_n$ and Lemma C.6. The $I$-channel controls local propagation within components; the $J$-channel couples to the global / mean direction and injects dense rank-one terms. In this section, we first shift to a micro-level perspective, focusing on the effects of individual samples (graphs), and then draw connection to how the training distribution determines the model's eventual behavior (algorithmic vs. heuristic, §4.3).

We decompose the single-sample loss $\mathcal{L}_G(\Theta) := \mathcal{L}(\mathsf{TF}_\Theta^L(A); R)$ and examine directional derivatives at a fixed $\Theta$, with the link gradient $\partial\mathcal{L}/\partial Z = \alpha(1 - R/\phi_\epsilon(Z))$. Throughout, we say a pair $(i, j)$ is **saturated** if its per-pair loss gradient vanishes; for within-component pairs ($R_{i,j} = 1$) this is equivalent to $\phi_\epsilon(Z_{i,j}) = R_{i,j}$. We say a direction $\Delta$ is **active** over $(i, j)$ if the corresponding channel directive $D_{i,j} > 0$, where $D$ denotes $\frac{\partial Z}{\partial A_\ell}[\Delta]$ or $\frac{\partial Z}{\partial B_\ell}[\Delta]$ as appropriate.

Our first main result is the following Theorem, which intuitively claims two things:

- *(Within capacity) Small-diameter graphs "reward" the local $I$-channel and, if disconnected, penalizes the global $J$-channel if activated.*
- *(Beyond capacity) Large-diameter connected graphs demand a global shortcut: the $J$-channel is promoted, while the $I$-channel remains confined to short-range corrections.*

**Theorem C.9** (Per-sample pushes by diameter). *Fix a layer $\ell$ and nonnegative directions $\Delta_A, \Delta_B \geq 0$ for $A_\ell, B_\ell$, respectively. Assume $B_1 = \ldots = B_L = 0$.*

(i) *(Within capacity) If $\mathrm{diam}(G) \leq 3^L$, then $D_{A_\ell}\mathcal{L}_G(\Theta)[\Delta_A] \leq 0$, with strict $< 0$ whenever $\Delta_A$ is active on at least one unsaturated within-component pair. If, in addition, $G$ is disconnected, then $D_{B_\ell}\mathcal{L}_G(\Theta)[\Delta_B] > 0$ if both of the following hold: $\Delta_B$ is active at least one cross-component pair, and cross component penalty dominates within-component reward (cf. equation 9).*

(ii) *(Beyond capacity) If $\mathrm{diam}(G) > 3^L$ and $G$ is connected, then we have $D_{A_\ell}\mathcal{L}_G(\Theta)[\Delta_A] \leq 0$ where only within-capacity pairs can contribute, and $D_{B_\ell}\mathcal{L}_G(\Theta)[\Delta_B] < 0$ for $\Delta_B$ that is active on at least one unsaturated pair.*

To prove this Theorem, we split the argument into the following four lemmas, each isolating one ingredient of the dynamics. Firstly, Lemma C.10 shows that the local $I$-channel is monotone: any nonnegative $A_\ell$ cannot increase the loss and is strictly helpful on unsaturated within-component pairs. This lets us treat local corrections as "harmless," while Lemma C.11 analyze the sign of the global $J$-channel (connected vs. disconnected), and Lemma C.12 determines which pairs are ever affected when $B = 0$. Together, they yield the two cases in Theorem C.9.

**Lemma C.10** (Local channel always helps). *Assume $B_1 = \ldots = B_L = 0$. For any graph $G$, any layer $\ell$, and any direction $\Delta \geq 0$ in the $I$-channel,*

$$D_{A_\ell}\mathcal{L}_G(\Theta)[\Delta] = \left\langle \frac{\partial\mathcal{L}}{\partial Z}, \frac{\partial Z}{\partial A_\ell}[\Delta] \right\rangle_F \leq 0, \tag{12}$$

*with strict inequality whenever there exists a within-component, unsaturated pair, on which $\Delta$ is active.*

*Proof.* From the block decomposition from equation 11, the $I$-channel contributes $\sum_{p,q} \Delta_{p,q} X_p X_q^\top$., which is block-diagonal with respect to the component partition. Hence $\frac{\partial Z}{\partial A_\ell}[\Delta]$ has support only on pairs $(i, j)$ in the same component. On those pairs, $R_{i,j} = 1$, and thus

$$\left(\frac{\partial\mathcal{L}}{\partial Z}\right)_{i,j} = \alpha \cdot \left(1 - \frac{1}{\phi_\epsilon(Z_{i,j})}\right) = -\alpha \cdot \frac{1 - \phi_\epsilon(Z_{i,j})}{\phi_\epsilon(Z_{i,j})} \leq 0,$$

with strict negativity whenever $\phi_\epsilon(Z_{i,j}) < 1$. Entrywise, nonnegativity of the forward map (Lemma C.6) gives $\frac{\partial Z}{\partial A_\ell}[\Delta] \geq 0$. Therefore the Frobenius inner product $\leq 0$, and $< 0$ under the stated conditions. $\square$

We now switch from the local $I$-channel to the global $J$-channel and will use that the forward sensitivity in the $J$-channel is entrywise nonnegative, so the sign of the directional derivative is controlled entirely by the per-pair loss gradient.

**Lemma C.11** (Global channel helps connected graphs and conditionally hurts disconnected graphs)**.** *Fix a layer $\ell$ and a nonnegative direction $\Delta \geq 0$ in the $J$-channel.*

*(i) If $G$ is connected, then $D_{B_\ell}\mathcal{L}_G(\Theta) \leq 0$, with strict $< 0$ whenever there exists an unsaturated pair $(i,j)$ (i.e., $\phi_\epsilon(Z_{i,j}) < 1$) on which $\Delta$ is active ($D_{i,j} > 0$).*

*(ii) If $G$ is disconnected, then*

$$D_{B_\ell}\mathcal{L}_G(\Theta)[\Delta] = \alpha \cdot \left[ \sum_{R_{i,j}=0} D_{i,j} - \sum_{R_{i,j}=1} \frac{1 - \phi_\epsilon(Z_{i,j})}{\phi_\epsilon(Z_{i,j})} D_{i,j} \right], \tag{13}$$

*hence $D_{B_\ell}\mathcal{L}_G(\Theta)[\Delta] \geq 0$ whenever the cross component mass penalty dominates the ratio-weighted within-component reward. Strict $> 0$ holds if the inequality is strict, and $\Delta$ is active on at least one cross pair.*

*Proof.* By the chain rule,

$$D_{B_\ell}\mathcal{L}_G(\Theta)[\Delta] = \left\langle \frac{\partial \mathcal{L}}{\partial Z}, \frac{\partial Z}{\partial B_\ell}[\Delta] \right\rangle_F = \left\langle \alpha \cdot \left( 1 - \frac{R}{\phi_\epsilon(Z)} \right), D \right\rangle_F. \tag{14}$$

By Lemma C.6 , $D \geq 0$ entrywise; moreover, $D_{i,j} > 0$ exactly on pairs where $\Delta$ is active.

(i) If $G$ is connected, then the reachability matrix $R$ is all-ones. Hence $\frac{\partial \mathcal{L}}{\partial Z} = -\alpha(1 - \phi_\epsilon(Z))/\phi_\epsilon(Z) \leq 0$ entrywise, with strict negativity whenever $\phi_\epsilon(Z_{i,j}) < 1$. Pairing with $D \geq 0$ and $D_{i,j} > 0$ on active pairs gives $D_{B_\ell}\mathcal{L}_G(\Theta)[\Delta] \leq 0$ and strict $< 0$ under the stated saturation / activation conditions.

(ii) If $G$ is disconnected, split equation 14 over $R_{i,j} = 0$ and $R_{i,j} = 1$ to obtain the displayed identity. Since $D \geq 0$, the stated dominance condition yields $\geq 0$. Strictness requires a cross pair with $D_{i,j} > 0$, holds exactly when $\Delta$ is active on at least one cross-component pair. □

We now show that when the global $J$-channel is disabled, the model can only light up within-capacity pairs. Note this is somewhat a converse to Theorem C.2, where a "good" model that only lights up within-capacity pairs necessarily have each $W_\ell[r, s]$ diagonal. The following Lemma isolates the role of the $J$-channel as the only "nontrivial" shortcut.

Recall from Definition 4.5: for a depth $L$ and a graph $G$ with adjacency matrix $A$, we call a pair $(i, j)$ **within capacity** if $[A^{3^L}]_{i,j} > 0$ and **beyond capacity** otherwise.

**Lemma C.12** ($I$-channel reaches within-capacity pairs; $J$-channel is the only dense shortcut)**.** *At any $\Theta$ with $B_1 = \ldots = B_L = 0$, the output satisfies*

$$Z_{i,j} > 0 \implies [A^{3^L}]_{i,j} > 0.$$

*Equivalently, beyond-capacity pairs receive no positive mass from the $I$-channel alone. In contrast, for any $\ell$ and any $\Delta \geq 0$ in the $J$-channel, $\frac{\partial Z}{\partial B_\ell}[\Delta] \geq 0$ and is strictly positive on active pairs by definition.*

*Proof.* Since $B_\ell = 0$ implies $h_{\ell-1}W_\ell h_{\ell-1}^\top = \sum_{p,q}(A_\ell)_{p,q}X_pX_q^\top$ from Lemma C.6, it is easy to see that they are block-diagonal w.r.t. connected components. Hence Lemma B.2 applies and support expands by at most a factor of 3 per layer, and only within-capacity pairs receive mass. The density statement follows from Lemma C.6: For any $\Delta \geq 0$ in the $J$-channel, we have $\frac{\partial Z}{\partial B_\ell} = \sum_{p,q} \Delta_{p,q}u_p u_q^\top u \geq 0$. The strict positiveness characterization follows directly from Lemma C.6. □

With the previous lemmas established, we can now assemble the per-sample sign rules. Intuitively, the $I$-channel makes only local corrections, never hurting the loss and only touching within-capacity pairs when $B = 0$, while the $J$-channel is the sole dense shortcut, helpful on connected graphs but penalized by cross-component pairs when the graph is disconnected.

*Proof of Theorem C.9.* Let $\ell, \Delta_A, \Delta_B$ be given as described. Set $D_A = \frac{\partial Z}{\partial A_\ell}[\Delta_A]$ and $D_B = \frac{\partial Z}{\partial B_\ell}[\Delta_B]$. Recall from chain rule

$$D_{(\cdot)}\mathcal{L}_G(\Theta)[\cdot] = \left\langle \frac{\partial \mathcal{L}}{\partial Z}, \frac{\partial Z}{\partial(\cdot)}[\cdot] \right\rangle_F = \alpha \cdot \left\langle 1 - \frac{R}{\phi_\epsilon(Z)}, \frac{\partial Z}{\partial(\cdot)}[\cdot] \right\rangle_F.$$

(i) (Within capacity) By Lemma C.10, for any $\Delta_A \geq 0$ the $I$-channel directional derivative is $\leq 0$, with strict inequality under the stated conditions. The result on disconnected graphs $G$ follows from Lemma C.11.

(ii) By Lemma C.12, with $B = 0$, only within-capacity pairs can be affected by the $I$-channel, so Lemma C.10 gives $D_{A_\ell}\mathcal{L}_G(\Theta)[\Delta_A] \leq 0$. Since $G$ is connected and $\mathrm{diam}(G) > 3^L$, there will be unsaturated pairs; then Lemma C.11(i) yields $D_{B_\ell}\mathcal{L}_G(\Theta)[\Delta_B] < 0$, as claimed. $\qquad\square$

**Remark C.13** (Population-level consequence under $\mathrm{ER}(n, p)$)**.** Fix a layer $\ell$ and nonnegative directions $\Delta_A, \Delta_B \geq 0$. Partition the graphs into $\mathcal{G}_0 = \{G : \mathrm{diam}(G) \leq 3^L\}$ and $\mathcal{G}_1 = \{G : \mathrm{diam}(G) > 3^L\}$. Writing the population directional derivatives as mixtures,

$$D_{B_\ell}\mathcal{R}(\Theta)[\Delta_B] = \mathbb{P}(\mathcal{G}_0)\mathbb{E}[D_{B_\ell}\mathcal{L}_G(\Theta)[\Delta_B] \mid G \in \mathcal{G}_0] + \mathbb{P}(\mathcal{G}_1)\mathbb{E}[D_{B_\ell}\mathcal{L}_G(\Theta)[\Delta_B] \mid G \in \mathcal{G}_1].$$
$$(15)$$

We claim the following on the population gradient.

(i) (Local) From Lemma C.10, once the global $J$-channel has been suppressed, the local $I$-channel is consistently promoted until saturation.

(ii) (Global) The population gradient along the global $J$-channel is an explicit mixture of two regimes: large, connected graphs beyond capacity that promote the $J$-channel, and small, disconnected graphs within capacity that suppress it whenever cross-component errors persist. Formally:

(ii.a) If $G$ is connected and $\mathrm{diam}(G) > 3^L$, then by Lemma C.12, every beyond-capacity pair has $Z_{ij} = 0$ while $R_{ij} = 1$. For those pairs, we have $\partial \mathcal{L}/\partial Z = -\alpha(1 - \phi_\epsilon(Z))/\phi_\epsilon(Z) < 0$. By Lemma C.6, the inner product $\langle \partial\mathcal{L}/\partial Z, \partial Z/\partial B_\ell[\Delta_B]\rangle_F < 0$ too. Integrating over all beyond-capacity, connected graphs yields

$$\mathbb{E}[D_{B_\ell}\mathcal{L}_G(\Theta)[\Delta_B] \mid G \in \mathcal{G}_1 \ \& \ G \text{ connected}] < 0. \qquad (16)$$

(ii.b) If $G$ is disconnected and $\mathrm{diam}(G) \leq 3^L$, then by Lemma C.11, $D_{B_\ell}\mathcal{L}_G(\Theta)[\Delta_B] \geq 0$ with strict $> 0$ if cross-component errors persist (the $\sum_{R=0} D$ term strictly dominates the $\sum_{R=1}(1 - \phi_\epsilon(Z))/\phi_\epsilon(Z) \cdot D$ term), and if $\Delta_B$ is active on cross pairs (i.e. $D_{ij}^{(B)} > 0$ for some $R_{ij} = 0$). The latter holds by Lemma C.6 if $\Delta_B$ is active on at least one cross pair. Integrating thus yields

$$\mathbb{E}\big[D_{B_\ell}\mathcal{L}_G(\Theta)[\Delta_B] \,\big|\, G \in \mathcal{G}_0 \ \& \ G \text{ disconnected}\big] \ \geq \ 0, \qquad (17)$$

and strictly $> 0$ provided the two additional assumptions above.

# D    ADDITIONAL EXPERIMENT DETAILS AND RESULTS

## D.1    EXPERIMENT DETAILS

**Standard Transformers.** When training 2-layer standard Transformers, we adopt the implementation from RoBERTa (Liu et al., 2019) with single-head per-layer and using normalized ReLU activation function as defined in Definition A.1. We use a hidden dimension of $d = 512$ to make sure the hidden size is not the blocker for expressivity. We trained on 1 Billion ER graphs with a batch size of 1000 and $10^6$ steps. Each graph is only seen by the model once to resembling the training regime of modern LLMs. We note that although 1 billion graphs sounds a lot but with $n = 20$ nodes, this is far from enumerating all possible graphs: there can be $2^{\binom{n}{2}}$ graphs if we don't consider graph isomorphism. When $n = 20$, this is about more than $10^{57}$ graphs in total, and 1 billion ($10^9$) is only a very small number of training instances. We train with AdamW optimizer with a learning rate of `1e-4` and weight decay of `1e-4` and a cosine learning rate decay.

**Disentangle Transformers.** For 1-layer disentangle transformers in Section 5, we train on a fixed set 4096 i.i.d. samples of $\mathsf{ER}(n = 8)$ graphs and running standard *Gradient Descent* without any mini-batching. In this case, we have a learning rate of $0.1$ with cosine learning rate decay. For 2-layer disentangled Transformers, we train on the same set of 1 billion number of $\mathsf{ER}(n = 20)$ graphs as with standard Transformers. For 3-layer models, we train on 1 billion number of $\mathsf{ER}(n = 64)$ graphs. Both 2- and 3-layer models are trained with AdamW with a learning rate of `1e-3`. We would like to note that the hidden dimensions $d_\ell$ of disentangle Transformers are fixed to be $d_\ell = 2^\ell n$ rather than a hyper-parameter (see Definition 4.1).

**Computing Energy Share of $I_n/J_n$ Channels.** In the experiments on 1-Layer disentangled Transformers, we compute energy shares of the $A \otimes I_n$ and $B \otimes J_n$ within $\|W\|_F^2$. Here is the formalized versions. We consider the noisy decomposition $W = \hat{A} \otimes I_n + \hat{B} \otimes J_n + W_\epsilon$, where $W_\epsilon$ is the projection error term. We define Frobenius-norm energy share on the $I_n$ channel as

$$\mathsf{EnergyShare}(\hat{A} \otimes I_n, W) = \frac{\langle W, \hat{A} \otimes I_n \rangle}{\|W\|_F^2} = \frac{\|\hat{A} \otimes I_n\|_F^2 + \langle \hat{A} \otimes I_n, \hat{B} \otimes J_n \rangle + \langle \hat{A} \otimes I_n, W_\epsilon \rangle}{\|W\|_F^2},$$

and by symmetry, the $J_n$-channel share is

$$\mathsf{EnergyShare}(\hat{B} \otimes J_n, W) = \frac{\langle W, \hat{B} \otimes J_n \rangle}{\|W\|_F^2} = \frac{\|\hat{B} \otimes J_n\|_F^2 + \langle \hat{B} \otimes J_n, \hat{A} \otimes I_n \rangle + \langle \hat{B} \otimes J_n, W_\epsilon \rangle}{\|W\|_F^2}.$$

This is a well-designed quantity because if you expand $\|W\|_F^2$ you obtain $\langle W, \hat{A} \otimes I_n + \hat{B} \otimes J_n + W_\epsilon \rangle$, and the $I/J$-channels' energy shares will sum to one when the projection error $W_\epsilon$ converges to zero.

## D.2    TRAINING DYNAMICS OF DISENTANGLED AND STANDARD TRANSFORMERS

In Figure 7, we show the training dynamics of a 3-Layer disentangled Transformer. In Figure 8, we show the learned weights by disentangled transformers.

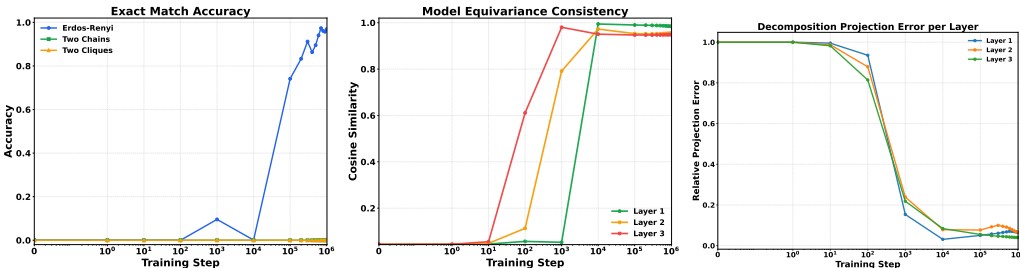

Figure 7: We plot the model behavior of a 3-Layer Disentangled Transformer model trained on $\mathsf{ER}(n = 64)$ graphs. They also quickly pick up almost *layer-wise equivariant* properties (measured by Eqn. 6). All layers show very small projection error onto the $A \otimes I_n + B \otimes J_n$ decomposition, resonating our theoretical claims in Theorem 4.6.

In Figure 8, we show that the trained 2-layer and 3-layer converge to weight spaces $W_\ell = A_\ell \otimes I_n + B_\ell \otimes J_n$ in the particular form echoing Theorem 4.6.

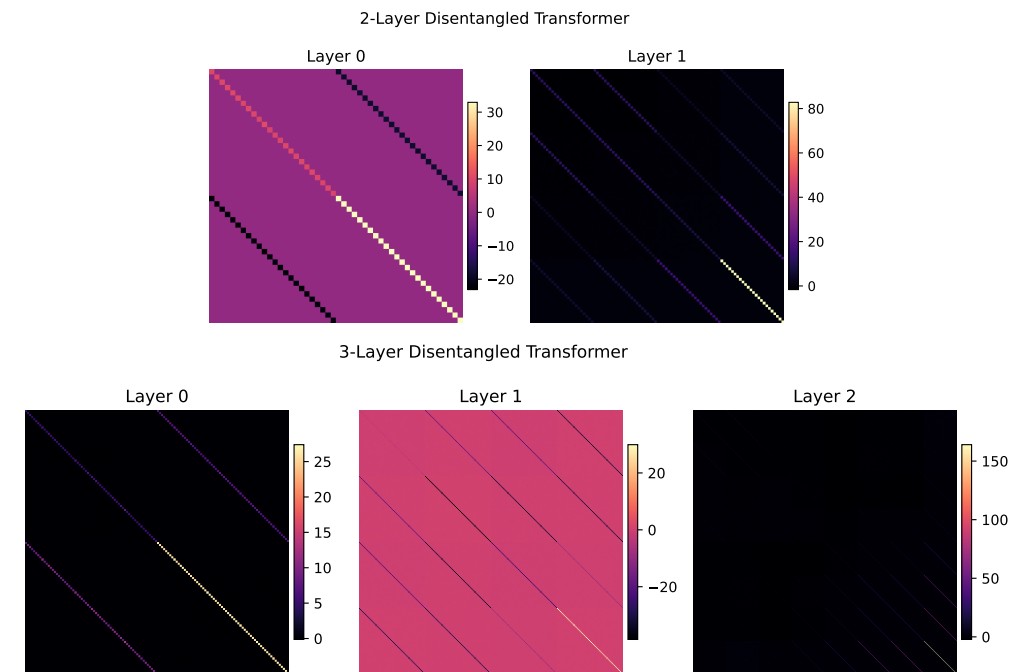

Figure 8: Here we visualize the weights $W_\ell$ learned by a 2-Layer and 3-Layer disentangled Transformer respectively. All models are randomly initialized **without** any restriction on parameterization. Resonating Theorem 4.6, they all converge to a form of $W_\ell = A_\ell \otimes I_n + B_\ell \otimes J_n$.

In Figure 9, we show that the capacity theorems (Theorem 4.4) also transfer to standard 2-layer Transformer models.

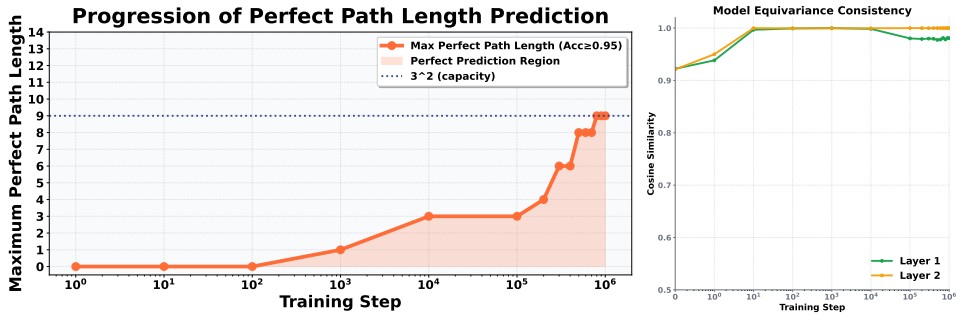

Figure 9: (**left**) Standard Transformers models studied in §3.3 also hit its capacity wall at $3^L$, showing that our theoretical results transfer beyond the theoretical simplification of disentangled transformers. (**right**) Standard Transformer models also learn an almost layer-wise equivariant solution measured by Eqn. 6.

## D.3 SCALING EFFECTS OF DIAMETER AND CAPACITY

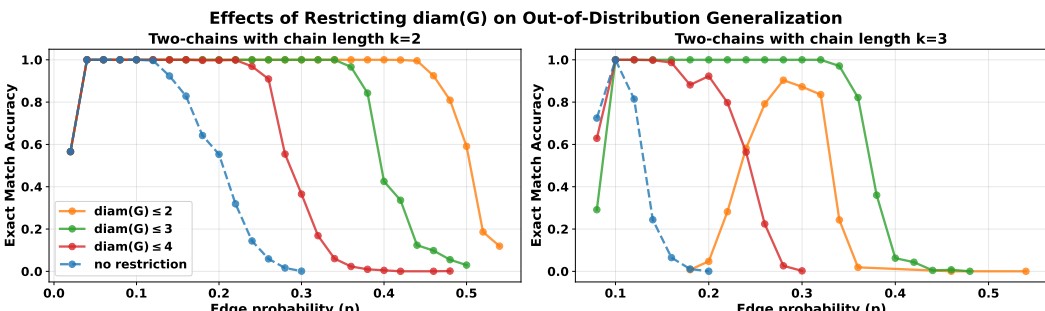

Figure 10: With 1-layer disentangled transformers with capacity $\mathsf{Cap} = 3$ following Theorem 4.4, we vary $d$ such that we restrict our training graphs to have $\mathrm{diam}(G) \leq d$. We also vary the edge probability of our training distribution $\mathsf{ER}(n = 8, p = \cdot)$ for generality. We test on $2\mathsf{Chain}(n = 8, k = \cdot)$ graphs with $k = 2$ or $3$ and show the exact match accuracy on configurations where the accuracy is non-zero for readability. We find if the training $d \leq \mathsf{Cap}$, models still learns the algorithmic solution up to problem size $d$ (see $d = 2, k = 2$ case on the left in orange) but *fails to length generalize* (see $d = 2, k = 3$ in orange on the right). On the other hand, if the training $d > \mathsf{Cap}$, model struggles to learn the algorithmic solution (see $d = 4$ cases in red on both $k = 2$ or 3). The best case overall is when setting $d = \mathsf{Cap}$, i.e., preventing the model from seeing beyond-capacity samples but still preserving at-capacity samples for better generalization. As shown in the green lines, with $d = 3$, model achieves balanced testing accuracy on both $k = 2$ and 3.

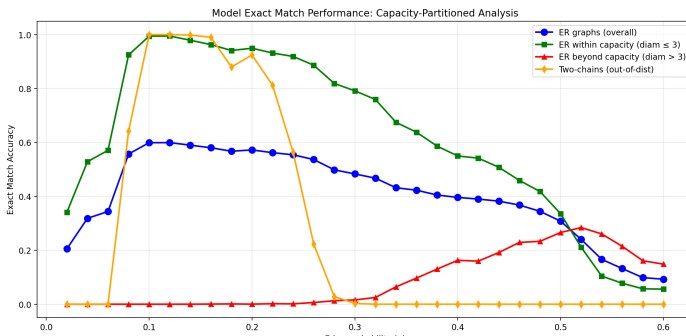

(a) When training 1-layer disentangled Transformers, instead of restricting training graphs to have diameter at most 3, we restrict $\mathrm{diam}(\mathbf{G}) \leq \mathbf{2}$ and varying the edge probability in $\mathsf{ER}(n = 8, p = p)$ training distribution. When measured by exact match accuracy, restricting $\mathrm{diam}(G) \leq 2$ make the models unable to generalize as well, indicating the importance of **at-capacity graphs** $(\mathrm{diam}(G) = 3)$

(b) When restricting $\mathrm{diam}(\mathbf{G}) \leq \mathbf{3}$, with reasonable $p \in [0.1, 0.32]$, 1-layer disentangled transformer can learn the algorithmic channel.

(c) When restricting $\mathrm{diam}(\mathbf{G}) \leq \mathbf{4}$, allowing some beyond-capacity graphs, 1-layer disentangled transformer struggle to learn the algorithmic channel, and starts to rely on the heuristic $J_n$-channel to make predictions on beyond-capacity graphs (red lines).

Figure 11: Effects of **at-capacity graphs** $(\mathrm{diam}(G) = 3^L)$ for $L = 1$. Without at-capacity graphs, models struggle to learn the algorithmic solution. With beyond-capacity graphs, models weight too much on heuristics. In short, models not only need most graphs within capacity and but also require at-capacity graphs to learn algorithms over heuristics.