# OpenReview forum: "When Do Transformers Learn Heuristics for Graph Connectivity?"
_ICLR.cc/2026/Conference — ICLR 2026 Conference Desk Rejected Submission_

### Official Review · Reviewer_E2Xj · 2025-10-19

**Soundness:** 3
**Presentation:** 2
**Contribution:** 2
**Rating:** 2
**Confidence:** 4

**Summary:**

This paper analyzes the capacity of transformers to generalize out of distribution on a toy task. Specifically, the paper analyzes the problem of predicting the connectivity matrix of a graph by supervising a transformer on a training set with ground-truth solutions. The paper aims to explain when the transformer learns an actual algorithm for solving this task, which can hence be transferred to any test distribution, and when the learned solution is overfitted to the training distribution. The analysis is based on a specialized transformer architecture, tailored to this task.

**Strengths:**

Analyzing the capabilities of neural networks through toy tasks is a valid approach, often providing intuition that extends to more general cases. The paper is well-written, and the theoretical derivations appear to be sound.

**Weaknesses:**

Some of the main results in this paper are interpreted in a misleading way. Specifically, one of the main contributions that the authors claim to have made is:

**If the capacity is not exceeded in the training set, then training will lead the transformer to learn the right algorithmic solution of the problem.**

This is claimed often in the paper. However, when reading the actual theoretical construction, what was proven is much weaker. An appropriate interpretation of what is proven is:

**The analysis suggests that if the transformer already implements the algorithmic solution, then training will not change this.**

The authors should delete all of the instances of the misleading claim from their paper, and instead write the more appropriate and modest interpretation of their theoretical contribution.

In more details:

The assumptions in Section 4.3 beat the whole point of this section. You want to show that if the capacity is not exceeded, then the training will lead the transformer to learn the right algorithmic solution of the problem, namely, to implement powers of the adjacency matrix. However, instead of proving this, in lines 276 – 278 you basically ASSUME that the transformer implements powers of the adjacency. More accurately, you assume something as strong, namely, that the transformer predicts the support of the powers of the adjacency, which is what you need to implement the algorithmic solution.

Then, starting from line 297, the authors actually assume that the transformer exactly implements the algorithmic solution. It is assumed that one channel of the transformer implements the algorithm, and another channel does not. The analysis only shows that for data under the capacity, the channel with the algorithmic solution dominates. However, this is a very weak result. The actual question that the authors set out to answer is whether the transformer can **learn** the algorithmic solution. This question is not answered by the analysis.  Instead, the analysis shows that under strict conditions, if the transformer already implements the algorithmic solution, training will not ruin this. Hence, the analysis in this section is interpreted in a misleading way. The authors should not claim to have solved the aforementioned problem (proving that below the capacity, transformers learn the algorithmic solution).  The authors should delete any such claim from the whole paper. For example, line 475–478 should be deleted. Instead, the authors should write that their analysis suggests that if the transformer already implements the algorithmic solution, then training will not change this.

Another point is that the paper does not discuss what the special toy task can teach us about the capacity of general transformers on general tasks to avoid overfitting by learning heuristics.


Minor comment:

Line 662: change the index h to another notation, as h already represents the features.

**Questions:**

See Weaknesses

---

> ### Author Response · Authors · 2025-11-21
>
> ### __Clarification on Our Theoretical Results.__
> We thank the reviewer for their close reading of our theoretical derivation. We respectfully __disagree__ that our theoretical results only show that “if the transformer already implements the algorithmic solution, training will not ruin this.”
>
> The main critique appears to stem from the impression that the training dynamics analysis assumes “the transformer [already] implements powers of the adjacency.” We believe this __misunderstanding__ arises from the placement of Theorem C.2 (line 278) versus Theorem 4.6 (line 283). Theorem C.2 characterizes the destination (what a “perfect” model should look like), whereas Section 4.3 analyzes the trajectory driven by gradients.
>
> Throughout Section 4.3, we (theoretically and empirically) assume a fixed parameter space (the “Kronecker algebra” of $\{I_n, J_n\}$), but we do not assume a solution; the justifications are summarized below. Theorem C.2 plays no role throughout this Section.
>
> - (Theoretical) At the beginning of dynamics analysis, the only assumption we need is that $W$ lies in the “Kronecker algebra” of $\{I_n, J_n\}$. This is given by Theorem 4.6: if a model is layerwise permutation equivariant, then its weights must satisfy the decomposition $W = A \otimes I_n + B  \otimes J_n$ for some $A, B$.  At this point, we do not care what $A,B$ are, nor do we know (or assume) the roles of  $I_n$ and $J_n$.  In other words, we rely on this parameterization, but not the solution itself, for section 4.3.
>
> - (Empirical) Figures 3 and 8 empirically validate that Disentangled Transformers naturally converge to this parameterization. Therefore, the $\{I_n, J_n\}$ parameterization of $W$ is both theoretically and empirically supported.
>
> We recognize that the placing theorem C.2 (which characterizes the optimal solution) at the beginning of the section may have been distracting. We updated the manuscript where we restructured the opening of Section 4.3 to clearly separate the “characterization of the solution” from the “analysis of training dynamics.” We will explicitly clarify in the text that the dynamics analysis assumes only the parameterization provided by Theorem 4.6, not in the optimal weights as in C.2. Once again, we appreciate the reviewer for pointing out the structural ambiguity in Section 4.3!
>
>
> ### __On generality to broader tasks.__
> Regarding the reviewer's question on broader tasks: Our study identifies a fundamental tension between compositional mechanisms (algorithms) and global broadcasting (heuristics). This insight extends to any dynamic programming task expressible via matrix operations over semi-rings (e.g., shortest path over the $(\min, +)$ semi-ring, where heuristics correspond to hop-counts). Practically, this provides a concrete lesson for training latent reasoning models (e.g., Coconut [Hao et al., 2024]): if a reasoning step requires a logical jump exceeding the architecture's single-pass capacity, the model will inevitably revert to heuristics. Thus, "capacity" acts as a strict constraint on the granularity of chain-of-thought data.
>
> Hao et. al. Training Large Language Models to Reason in a Continuous Latent Space

---

### Official Review · Reviewer_1s5u · 2025-10-27

**Soundness:** 3
**Presentation:** 2
**Contribution:** 4
**Rating:** 6
**Confidence:** 3

**Summary:**

Authors propose a study to detect whether (disentangled) transformers learn algorithms rather than shortcuts (i.e. brittle heuristics vs. correct algorithmic solutions). They design specific stress tests based on graph connectivity, because it is a fundamental algorithmic problem which admits simple heuristics and has a complexity landscape tied to Transformers depth.

The authors
(i) prove that a $L$-layer model can solve graphs with diameter $\leq 3L$ by effectively computing powers of the adjacency matrix; beyond this, it must fail on some instances.
(ii) They characterize training dynamics via a *two-channel decomposition* (algorithmic ($A \otimes I$) vs. heuristic ($B \otimes J$), the latter encoding degree-based shortcuts), showing which channel wins depends on the *fraction of beyond-capacity graphs* in the training distribution.
(iii) They propose a data lever training that suppresses heuristics and yields improved OOD generalization. They show that the effect *transfers to standard Transformers* as well as the disentangled model.

Experiments on synthetic ER graphs and OOD "two-chains/cliques" support the theory.

**Strengths:**

* The theoretical results are clear and well-defined. The *expressivity theorem* and *tight $3L$ capacity* (Theorems 4.3-4.4) present their main points clearly, including important assumptions like self-loop augmentation, depth L, and non-negativity of the capacity upper limit. These results come with proof sketches and detailed appendices.

* The *layerwise permutation-equivariant parameterization* (Theorem 4.6) provides a clear method for understanding the difference between *algorithmic* and *heuristic* channels, which helps analyze *gradient dynamics* (Theorems C.5/C.9).

* Figures from the experiments support three main findings: (i) the *$3L$ capacity breakpoint*, (ii) the *two-phase training behavior*, and (iii) the *data-lever mitigation*. They also show the *transferability* to standard Transformers. Although some assumptions, such as *non-negativity* and *equivariance form*, are idealized, the authors talk about these issues directly. The predictions they make also align well with the results from experiments conducted *without* enforcing those constraints. Overall, the claims are *well-supported* and consistent with each other.

* The *data lever* is simple (restrict to *within-capacity* diameter) yet *effective*; it *transfers* to standard Transformers and improves OOD generalization on 'two-chains/cliques'

* Careful delineation of assumptions; synthetic testbed isolates the phenomenon; reproducibility statement and code availability are noted.

In sum, the work offers a novel law for connectivity (diameter $\leq 3L$), a mechanistic decomposition of training into algorithmic vs. heuristic channels with predictive training-dynamics theory, and a *simple, effective data intervention that enhances OOD generalization even for standard Transformers. Given the community's interest in *algorithmic reasoning*, *shortcut learning*, and *mechanistic interpretability*, these contributions are *timely* and *impactful* for ICLR.

**Weaknesses:**

* The paper is well-structured overall, but the narrative is dense; adding a short executive overview at the end of the Introduction (5-6 sentences: problem -> $3L$ capacity result -> why tight -> training-dynamics story -> data lever -> empirical scope) would help orient readers. In addition, I suggest collecting all assumptions (self-loops, non-negativity, permutation-equivariance, depth L, task definition) in a single callout box with 1-line justifications and pointers to where they're relaxed/validated would increase the clarity and the scope of the author's contributions. I'd also suggest considering a simple running example (small 6-8 node graph) reused across sections to illustrate: (i) layer reach vs. diameter, (ii) algorithmic vs. heuristic channels, (iii) effect of the data lever.
* Experiments rely on Erdős-Rényi graphs and a couple of OOD structures (two chains/cliques). Additional graph families (grids, small-world, scale-free, trees, expanders, barbell/lollipop) and directed/weighted variants would better probe the generality of the theory and the data lever.
* In realistic settings, one may *not know capacity* (e.g., mixed tasks, unknown diameters) or be unable to *filter by diameter*. A discussion/experiment on *approximate proxies* (e.g., curriculum by *graph diameter estimates*, path-length sampling, synthetic augmentation) would make the prescription more deployable.
*  Since connectivity is a classic *message-passing* problem, comparisons to *GNNs* (e.g., MPNNs with sufficient layers), *RNNs*, or *explicit DP modules* would contextualize the Transformer's behavior and whether the data lever similarly helps/hurts those models.
* Limited analysis of *n-scaling* and sensitivity to *p* in ER(n,p); more *systematic sweeps* could test how the within/beyond-capacity mixture threshold ( $\rho^\star$ ) moves with size/density and training budget.

**Questions:**

1. How sensitive is the *$3L$* upper bound to the *non-negativity* requirement? Do you observe empirical counterexamples to the $3L$ law when weights are unconstrained and trained longer/larger?
2. If one cannot compute diameters exactly at scale, which *practical proxies* of the diameter (e.g., layered BFS samples, effective resistance bounds, eccentricity estimates) suffice to approximate the within-capacity filter? Any experiments with *noisy diameter filters*?
3. Would the *two-channel* story extend to other algorithmic tasks (e.g., *shortest paths/reachability variations*, *cycle detection*, *parity/motif counting*)? Are there analogous *rank-1 broadcast* heuristic channels for these problems?
4. How does performance change on *scale-free* and *small-world* graphs where *degree distributions* or *clustering* differ markedly from ER? Does the heuristic channel bias shift?

I'm happy to revise my score upward if the authors (1) extend experiments to additional graph families and directed/weighted settings, (2) include a curriculum/proxy version of the data lever, and (3) add comparative baselines (e.g., MPNNs, GNNs) and a more systematic scaling analysis. Nonetheless, the work is solid and insightful, and I recommend it for acceptance.

---

> ### Author Response · Authors · 2025-11-21
>
> We thank the reviewer for their careful reading and strong evaluation, particularly for describing our work as "timely and impactful" and recognizing the "excellent" contribution. We address the suggestions below, clarifying why we maintain our specific scope while adopting your excellent presentation suggestions.
>
> ### __(W2, W4, Q4) Graph families and baselines.__
> The reviewer suggests extending experiments to other families (like MPNNs). While we appreciate the suggestion for breadth, we respectfully argue that adding MPNNs would conflate two different types of failure modes:
> - MPNNs (Structural): An $L$-layer MPNN has a receptive field of exactly $L$. It is structurally impossible for it to solve connectivity on graphs with diameter $>L$. Its failure is a hard hardware limitation, not a learning preference.
> - Transformers (Dynamic): Our work highlights a more subtle phenomenon. An $L$-layer Transformer can solve diameters up to $3^L$ (Theorem 4.3), which is exponentially larger than the MPNN. However, even when the problem is solvable ($L < \text{diam} \le 3^L$), the Transformer chooses to use a heuristic due to training dynamics.
>
> Therefore, comparing them would not illuminate the "Algorithm vs. Heuristic" tension, as the MPNN has no algorithmic option for large diameters. We focus on Erdos Renyi graphs because they allow us to smoothly vary the diameter to probe the precise $3^L$ boundary (Figure 5, 10 and 11).
>
> ### __(W3, Q2) Practicality and proxies__
> We agree that exact diameter calculation can be expensive at scale. However, we argue that a "noisy proxy" is sufficient:
> - Robustness (Figure 5): We explicitly show that the "Data Lever" works even if we do not filter all beyond-capacity graphs. As long as the proportion of hard graphs is reduced, the algorithmic channel dominates.
> - Proxies: Therefore, a cheap proxy (like randomized BFS from a few source nodes) effectively acts as a "noisy filter." Since our method is robust to noise, it is robust to the approximation errors inherent in using proxies.
>
> ### __Q1. Sensitivity to assumptions__
> > How sensitive is the $3^L$ upper bound to the non-negativity requirement?
>
> Our theory is highly predictive of standard models. As shown in Figure 9, Standard Transformers (trained with AdamW, __no non-negativity constraints__) hit the exact same $3^L$ capacity wall as the theoretical model. Furthermore, Figures 3 and 8 show that unconstrained models spontaneously converge toward the equivariant parameterization (Algorithm vs. Heuristic channels) predicted by our theory.
>
> ### __W1. Presentation and Writing__
> We appreciate the suggestions regarding the narrative structure. We believe our current Introduction (in the "Our Contributions" block) and Problem Setup (Section 3) already clearly define the scope. Our current "Our Contributions" block indeed already followed the reviewer’s suggested roadmap (problem $\rightarrow$ capacity result $\rightarrow$ why tight $\rightarrow$ training-dynamics story $\rightarrow$ data lever $\rightarrow$ empirical scope). We thank the reviewer for synergizing with us on presentation. Additionally, we have revised the presentation of Section 4.3 to ensure the assumptions are explicitly highlighted.

---

> > ### Comment · Reviewer_1s5u · 2025-11-24
> >
> > I thank the authors for their thoughtful and detailed rebuttal.
> >
> > On the theory side, the clarifications around the non-negativity assumption and the pointers to Figures 3, 8, and 9 alleviate my concern in Q1: I am satisfied that the proposed (L^2) capacity law is empirically robust in their unconstrained Transformer setting. I also appreciate the discussion of practicality and proxies (W3/Q2), emphasizing that the data lever remains effective when the diameter filter is noisy and suggesting that randomized BFS-style proxies make the prescription feel more deployable, although a small explicit ablation with an approximate proxy would still have been very helpful.
> >
> > Regarding experimental breadth and baselines (W2, W4, Q4), the distinction between the "algorithm vs. heuristic" phenomenon in Transformers and other architectures (MPNNs) is helpful and clarifies the intended scope. That said, there is a lack of additional graph families (e.g., small-world/scale-free).
> >
> > On presentation (W1), the authors note that the "Our Contributions" block already follows the suggested roadmap and that Section 4.3 has been revised to explicitly highlight assumptions, which goes some way toward addressing my clarity concerns.
> >
> > Overall, the rebuttal strengthens my confidence that the main theoretical and mechanistic claims are sound and that the data-lever insight is robust, even if some of my requests for broader empirical validation and baselines remain unmet. I therefore keep my original score, which is already positive, and a recommendation for acceptance.

---

### Official Review · Reviewer_VnQA · 2025-10-29

**Soundness:** 2
**Presentation:** 2
**Contribution:** 2
**Rating:** 2
**Confidence:** 4

**Summary:**

This paper studies the computational capabilities of Transformers without  CoTs. Specifically, the authors focus on the problem of graph connectivity, and analyze whether Transformer can learn to decide connectivity properties. They prove that constant-depth Transformers fail to solve connectivity, while log-depth Transformers can succeed. The paper also provides empirical evidence supporting these theoretical separations.

**Strengths:**

- Though [1] has already proven that  log-depth Transformers can solves graph connectivity, this paper gives a more fine-grained analysis.
- The paper is clearly written overall.
- The authors have conducted experiments to support the theoretical findings.

[1] William Merrill and Ashish Sabharwal. A little depth goes a long way: The expressive power of log-depth transformers.

**Weaknesses:**

1. The paper investigates the computational power of Transformers **without CoT**. However, as the authors themselves mentioned, it is already well known that Transformers without CoT are severely limited in expressive power. Moreover, these limitations can be easily removed by using CoTs: theoretically, Transformers with CoTs are known to be Turing complete, e.g. [2,3]; empirically, almost all practical Transformers make extensive use of CoTs. Therefore, the question explored here feels of limited relevance to either current practice or deeper theoretical understanding.

2. The paper focuses on a single specific task, namely graph connectivity. Prior works [1] have already shown that log-depth Transformers can solve graph connectivity. As such, the contribution mainly confirms previously understood phenomena rather than discovering fundamentally new behavior. It is not clear whether the insights here can be generalized to broader classes of graph problems. This makes the work more of a case study than a general theoretical advance.

[2] William Merrill and Ashish Sabharwal. The Expressive Power of Transformers with Chain of Thought.

[3] Qian Li and Yuyi Wang. Constant bit-size transformers are Turing complete.

**Questions:**

Can the analysis be extended to provide insights into whether current training procedures lead Transformers (with CoT) to simulate broader classes of algorithms?

---

> ### Author Response · Authors · 2025-11-21
>
> We thank the reviewer for their evaluation and for finding our analysis fine-grained and our writing clear. However, we respectfully believe there are two fundamental misunderstandings regarding the goal of our work compared to prior literature (e.g., [1]) and its relevance to Chain-of-Thought (CoT).
>
> ### __W2. Novelty vs. prior work: existence is not learning.__
> The reviewer summarizes our work as confirming that "log-depth Transformers can succeed," citing [1]. If our goal were merely to show that a solution exists, the reviewer would be correct. However, our focus is learnability.
>
> - [1] proves expressivity: They show a weight setting exists to solve connectivity. They do not determine if gradient descent can find it.
> - We analyze dynamics: We ask __why__ models often fail to find this solution despite having the expressivity. We derive an exact, non-asymptotic bound ($3^L$) and prove that when training data exceeds this instance-dependent capacity, gradients actively suppress the algorithmic channel in favor of heuristics. This yields a prescriptive intervention (the "data lever") which [1]’s asymptotic existence proofs cannot provide.
>
> In short, [1] tells us a solution is possible. Our theory tells us how to actually find it using Gradient Descent. This is a distinct and practical contribution that [1] cannot provide.
>
> ### __W1. Relevance of standard transformers in a CoT era.__
> We strongly disagree that CoT renders the analysis of the standard forward pass irrelevant.
>
> - The atomic unit of reasoning: A Chain of Thought is simply a sequence of forward passes. If the atomic forward pass prefers statistical heuristics over correct logical operations due to inductive bias, the resulting chain will be built on shortcuts or hallucinations.
>
> - Turing completeness $\neq$ learnability: While CoT makes Transformers Turing complete in theory, this does not guarantee that they can learn algorithms via gradient descent. Understanding the inductive bias of the base architecture is a prerequisite for building reliable CoT systems.
>
>
>
> ### __Q1. Implications for CoT training.__
> Regarding the reviewer's question on broader algorithms: Yes, our findings provide direct guidelines for CoT design. Our theory implies that training data must be granular enough so that each individual reasoning step falls within the capacity of the model's single forward pass (i.e., the "logical jump" must be $\le 3^L$). If a step in the reasoning chain exceeds this limit, our dynamics analysis predicts the model will revert to heuristics. This suggests that latent reasoning models (like Coconut [4]) must allocate sufficient steps to ensure no single transition exceeds the algorithmic limit of the architecture.
>
>
> [4] Hao et. al. Training Large Language Models to Reason in a Continuous Latent Space

---

> > ### Comment · Reviewer_VnQA · 2025-11-26
> >
> > Thank you for clarifying the contribution on training dynamics. I am not opposed to acceptance if the other reviewers consider the learnability results strong enough, but I do not find the expressiveness aspect sufficiently interesting on their own.

---

### Official Review · Reviewer_7GWi · 2025-10-31

**Soundness:** 3
**Presentation:** 3
**Contribution:** 2
**Rating:** 4
**Confidence:** 4

**Summary:**

This paper investigates why Transformers  fail to learn true algorithmic reasoning and instead rely on unstable heuristics. Authors focus on  the graph connectivity task as a testbed, and introduce a simplified model called the disentangled Transformer and prove that an L-layer version can solve graph connectivity for graphs with diameters up to $3^L$ -- by effectively computing powers of the adjacency matrix.  When training data mostly include graphs within this “capacity”, the model learns the correct algorithm. However, when exposed to larger graphs beyond its capacity, it defaults to a heuristic based on node degrees. To validate this, authors provide experiments with data that is within the capacity of the model (with specific adjustments to the model) and observe that the true algorithm is emulated successfully.

**Strengths:**

- Theoretically explaining the algorithmic alignment capacity of Transformers is interesting and important.
- A tight bound for the *capacity* of Transformers for the graph connectivity problem.
- Experiments are carefully designed and validate the presented theory.
- The paper is generally well-written: it is easy to read and follow.

**Weaknesses:**

- The scope of the contribution is very limited: the result applies only to graph connectivity and to the disentangled Transformer architecture. Of course, graph connectivity may serve as a good task setting for testing algorithmic alignment, but eventually we are interested in designing better architectures that generalize out of distribution, for a variety of task settings. Clearly we cannot hope to cover every task, and every domain, but the current scope is too limited in my opinion.
- The technical construction proving $3^L$ capacity is straightforward, given the connection with the transitive closure. That being said, showing that this is also a matching lower bound requires a careful construction.
- Experimental setup is clearly tailored to the specific task setting, which again makes sense in the context of this paper, but it keeps us in this extremely limited scope.
- Authors seem to suggest that "expressivity" and the "model capacity" are two different things. This needs to be toned carefully in my opinion. In the end, these are all expressivity results, but they differ in the kinds of assumptions they make. Many of these expressivity results (e.g., universality results) are non-uniform and require to show: for any input, there exists a model parametrization that realizes the target function (i.e., we are allowed to change the parametrization based on the input size). This is clearly weaker than uniform results which require the existence of one model parametrization that captures the target function on any input. The setup authors study is somewhere in-between, because it is essentially saying that, there exists a  a model parametrization that captures the target function on any input with diameter $\leq3^L$. This is all to say that a more delicate discussion is needed when it comes to expressivity. It is clear that uniform expressivity typically correlates with algorithmic generalization whereas non-uniform results do not inform us much (and similar for asymptotic ones).

**Questions:**

- Is there a way of generalizing the scope of this study to capture a broader class of functions? The bounds may be different for each function of course, but the theorems could be generally applicable to derive bounds -- based on the diameter/size of the graphs and the nature of the functions considered.

---

> ### Author Response · Authors · 2025-11-21
>
> We thank the reviewer for their thoughtful evaluation, particularly for recognizing the tightness of our bounds ($3^L$), the rigorous experimental design, and the clarity of our writing. We address the feedback regarding scope and technical contributions below.
>
> ### __W1 & W3. The necessity of a controlled scope.__
> We respectfully argue that our constraints (Disentangled Transformers, graph connectivity) are necessary controls, not weaknesses. To rigorously study why models fail to generalize OOD, we needed a theoretical setting where the competition between "heuristic" and "algorithm" could be tracked exactly.
>
> - Task: We focused on Connectivity because it admits an exact, non-asymptotic threshold ($3^L$). This precision is the prerequisite for our training dynamics analysis; without it, we could not mathematically derive the "phase transition" where gradients flip from learning the algorithm to learning the heuristic (Theorems C.5 & C.9).
>
> - Model: We treat the Disentangled Transformer as a proxy to make these gradient dynamics tractable. Crucially, our findings transfer directly: Figure 9 shows Standard Transformers hitting the exact same $3^L$ capacity wall, and Figure 6 shows that our "Data Lever" mitigation strategy successfully improves their OOD generalization.
>
>
> ### __W2. Technical novelty: construction vs. impossibility.__
> We agree that the constructive result (Theorem 4.3) is straightforward. However, our primary theoretical innovation regarding capacity is the __impossibility__ result (Theorem 4.4). Proving that an $L$-layer model cannot solve diameters greater than $3^L$ is non-trivial; it requires a careful analysis of the residual stream's receptive field to prove that information strictly cannot propagate faster than base-3 exponentiation. This impossibility result is foundational, as it defines the precise set of "beyond-capacity" graphs that drive the model toward global heuristics.
>
> ### __W4. Clarifying expressivity vs. capacity.__
> We appreciate the nuance regarding expressivity. In our context, we distinguish these terms to highlight a specific learning dichotomy: we view __expressivity__ as a lower bound (Does a solution exist? Yes, Theorem 4.3), and __capacity__ as an instance-dependent upper bound (Does this input fit the receptive field? Only if diameter $\le 3^L$, Theorem 4.4). The failure modes we analyze arise not because the model lacks the expressivity to represent the algorithm, but because the training data contains instances that exceed the model's capacity.
>
> ### __Q1. Generalization to broader functions.__
> Regarding the generalizability of our study: Yes, our framework naturally extends beyond connectivity. Our "Algorithm vs. Heuristic" decomposition relies on the tension between a compositional channel (matrix multiplication) and a rank-1 broadcast channel (global statistics). This structure exists in the broad class of dynamic programming algorithms expressible via matrix operations over semi-rings. For example, in a shortest-path task over the $(\min, +)$ semi-ring, the "heuristic" channel would correspond to simple hop-counts rather than true edge-weight sums. We hypothesize that for any task in this class, "Capacity" is determined by the number of composition steps relative to depth, while the "Heuristic" is driven by global aggregation.

---

### Author Response · Authors · 2025-11-21
**General Response**

We thank the reviewers for their constructive feedback. To clarify the scope and novelty of our work, we summarize our central narrative below:

### __1. Motivation: Existence/Expressivity is not Learning__

Prior work (e.g., Merrill & Sabharwal) established that log-depth Transformers can express the connectivity algorithm. We address a fundamentally different question:

__Even if the model is expressive enough to solve the task, why does it often fail to learn the algorithm?__

*   Definition: We define "learning an algorithm" as acquiring a mechanism (matrix powering) that generalizes to Out-of-Distribution (OOD) graphs (e.g., 2-Chain and 2-Cliques) where heuristics fail.
*   Phenomenon: Standard training achieves perfect in-distribution accuracy but 0% OOD accuracy (Figure 1), proving that transformer models trained with gradient descent prefer brittle heuristics over the robust algorithm. Our goal is to mechanically explain and fix this preference.

### __2. Why the Exact $3^L$ Bound is Necessary__

To explain this failure, we derive an exact capacity limit, not just an asymptotic bound.
*   Theorems 4.3 (Expressivity, lower bound) & 4.4 (Capacity, upper bound): We prove an $L$-layer model can solve graphs with diameter $\le 3^L$, but must fail on graphs with diameter $> 3^L$.
*   Why this is important: This exact threshold allows us to define a rigorous dichotomy (Definition 4.5): partitioning training data into within-capacity and beyond-capacity sets. Without the exact $3^L$ boundary, the subsequent dynamics analysis would be impossible.

### __3. From Capacity to Training Dynamics__

We link this capacity limit directly to gradient descent behavior using a rigorous weight decomposition:
*   Theorem 4.6 (Two Channels): Weights split into an Algorithmic Channel ($A \otimes I$, matrix powering) and a Heuristic Channel ($B \otimes J$, degree shortcuts). The only assumption we made for this Theorem is that the model should be permutation equivariant – a common assumption for permutation equivariant structures such as graphs.
*   Theorems C.5 & C.9 (Dynamics): We prove a phase transition in the gradients:
    *   Within-Capacity data ($\le 3^L$) drives gradients toward the Algorithmic channel.
    *   Beyond-Capacity data ($> 3^L$) actively drives gradients toward the Heuristic channel.

### __4. Prescriptive Power__

This theory prescribes the "Data Lever": filtering data by diameter $\le 3^L$ suppresses heuristics. Figure 6 validates that this theoretical insight successfully transfers to Standard Transformers, enabling perfect OOD generalization. We also would like to empathize this prescriptive power is only given if we have an exact bound $3^L$ to define a dichotomy (Definition 4.5), where asymptomatic tight bounds given by Merrill & Sabharwal couldn't.

*William Merrill and Ashish Sabharwal. A little depth goes a long way: The expressive power of log-depth transformers.*

---

### Note · Program_Chairs · 2026-01-17
**Submission Desk Rejected by Program Chairs**

The following references in this submission do not refer to real documents and/or have major errors in bibliographic information:

     Yifei Yuan, Shiqi Cheng, Yutai Hou, Xu Sun, and Lei Li. Do llms overcome shortcut learning? an evaluation of shortcut robustness in large language models. In EMNLP, 2024. URL https: //aclanthology.org/2024.emnlp-main.679.pdf.
    Kuan-Chieh Kao, Shumin Deng, Fan Yang, Zheng Li, and et al. Can large language models solve complex math problems? a comprehensive assessment. In Findings of EMNLP, 2024. URL https://aclanthology.org/2024.findings-emnlp.980.pdf. 1
    Andrei Cosma, Viorel Andrei, Dan Istrate, and Roxana Istrate. How hard is this test set? nli characterization by dataset-driven heuristics. In EMNLP, 2024. URL https://aclanthology. org/2024.emnlp-main.175.pdf.